Manuscript prepared for Atmos. Meas. Tech.
with version 2015/04/24 7.83 Copernicus papers of the LATEX class copernicus.cls.
Date: 2 November 2017

# Temperature uniformity in the CERN CLOUD chamber

António Dias[1], Sebastian Ehrhart[1,a], Alexander Vogel[1,b], Christina Williamson[2,c], João Almeida[1,2], Jasper Kirkby[1,2], Serge Mathot[1], Samuel Mumford[1,d], and Antti Onnela[1]

[1]CERN, CH-1211 Geneva, Switzerland
[2]Goethe University Frankfurt, Institute for Atmospheric and Environmental Sciences, 60438 Frankfurt am Main, Germany
[a]Now at Max Planck Institute for Chemistry Atmospheric Chemistry Department Hahn-Meitner-Weg 1, 55128 Mainz, Germany
[b]Now at Paul Scherrer Institute, Aarebrücke, 5232 Villigen, Switzerland
[c]Now at Chemical Sciences Division, NOAA Earth System Research Laboratory, Boulder, CO and CIRES, University of Colorado, Boulder, CO
[d]Now at Kapitulink Lab, 476 Lomita Mall, Stanford University, Stanford, CA 94305-4045

*Correspondence to:* António Dias (amcbd89@gmail.com)

**Abstract.** The CLOUD (Cosmics Leaving OUtdoor Droplets) experiment at CERN is studying the nucleation and growth of aerosol particles under atmospheric conditions, and their activation into cloud droplets. A key feature of the CLOUD experiment is precise control of the experimental parameters. Temperature uniformity and stability in the chamber are important since many of the
5 processes under study are sensitive to temperature and also to contaminants that can be released from the stainless steel walls by upward temperature fluctuations. The air enclosed within the 26 m$^3$ CLOUD chamber is equipped with several arrays ("strings") of high precision, fast-response thermometers to measure its temperature. Here we present a study of the air temperature uniformity inside the CLOUD chamber under various experimental conditions. Measurements were performed
under calibration conditions and run conditions, which are distinguished by the flow rate of fresh air and trace gases entering the chamber: 20 l/min and up to 210 l/min, respectively. During steady-state calibration runs between -70 °C and +20 °C, the air temperature uniformity is better than ±0.06 °C in the radial direction and ±0.1 °C in the vertical direction. Larger non-uniformities are present during experimental runs, depending on the temperature control of the make-up air and trace gases
(since some trace gases require elevated temperatures until injection into the chamber). The temperature stability is ±0.04 °C over periods of several hours during either calibration or steady-state run conditions. During rapid adiabatic expansions to activate cloud droplets and ice particles, the chamber walls are up to 10 °C warmer than the enclosed air. This results in temperature differences of ±1.5 °C in the vertical direction and ±1 °C in the horizontal direction while the air returns to its
equilibrium temperature with time constant of about 200 s.

# 1 Introduction

The Intergovernmental Panel on Climate Change (IPCC) considers that the largest source of uncertainty in anthropogenic radiative forcing of the climate is due to increased aerosol since pre-industrial times, and its effect on clouds (Myhre et al., 2013). Most of the increased aerosol has resulted from anthropogenic precursor vapours that, after oxidation in the atmosphere, can form particles which may then grow to become new cloud condensation nuclei (CCN). By current estimates, more than half of all CCN originate from nucleation rather than being emitted directly into the atmosphere (Gordon et al., 2017), but the vapours and mechanisms responsible remain relatively poorly known.

Laboratory experiments for studying atmospheric processes such as new particle formation (NPF) under controlled conditions have generally used tank or flow reactors. Raes and Janssens (1986) used a flow reactor made of glass to study ion-induced nucleation at 22 °C. A similar setup was used by Ball et al. (1999) to study nucleation of sulfuric acid, water and ammonia. Large teflon chambers provide lower loss rates to the walls and allow correspondingly longer residence times, making it feasible to grow aerosol particles to larger sizes and at lower vapour concentrations (Cocker et al., 2001).

The CLOUD experiment at CERN (Kirkby et al., 2011) has achieved sufficient suppression of contaminants (Schnitzhofer et al., 2014) inside a large, 3 m diameter stainless steel chamber to allow controlled NPF and cloud activation experiments to be performed over the full range of tropospheric temperatures and trace gas concentrations. CLOUD has presented a series of measurements of atmospheric particle formation rates for different chemical systems. Theoretical considerations and early measurements (Lovejoy, 2004) indicated a strong temperature dependence for the nucleation rates of sulfuric acid particles. This has been confirmed by CLOUD in, for example, Kirkby et al. (2011) and Kürten et al. (2016).

Expansion chambers are used to study in-cloud processes such as the homogeneous freezing of super-cooled liquid droplets (Möhler et al., 2003) or cloud droplet chemistry (Jurányi et al., 2009). These experiments require the formation of cloud droplets or ice particles on CCN in the chamber. Clouds can be formed in the CLOUD chamber by adiabatic expansion and cooling of humid air. This allows cloud microphysics and aqueous-phase chemistry to be studied in the CLOUD chamber, for example, phase transitions of cloud particles (Nichman et al. (2016)), or aqueous phase oxidation of $SO_2$ in cloud droplets (Hoyle et al. (2016)).

To understand new particle formation rates in the troposphere and lower stratosphere requires measurements at their ambient temperatures. Moreover, experimental NPF studies require stable temperatures over periods of several hours and a near-homogeneous temperature distribution over a large-volume experimental vessel. In the case of cloud formation by adiabatic expansions, fast-response and accurate temperature sensors are required to measure quantities such as the homogeneous freezing temperature or the onset of cloud droplets and their later evaporation. Here we present a study of the temperature uniformity and accuracy achieved in the CLOUD chamber under a) ideal

conditions (with no deliberate additional heat) and b) operational conditions (where additional heat is introduced into the chamber by UV light and warm gases).

 ## 2    Chamber operation

The CLOUD chamber is a 26 m$^3$ cylindrical stainless steel vessel which is filled with humidified artificial air and chosen trace atmospheric vapours such as $O_3$ or $SO_2$. The ion concentrations inside the chamber can be precisely controlled over the full tropospheric range with a pion beam from the CERN Proton Synchrotron(Suller and Petit-Jean-Genaz, 1995). To ensure adequate mixing of the chamber, two large mixing fans operate at the top and bottom of the chamber, respectively (Voigtlander et al., 2012).

   The CLOUD chamber normally operates at 5 mbar above atmospheric pressure (the small excess ensures that no contaminant vapours enter the chamber through the sampling ports). However, for cloud formation experiments, the air in the chamber is first raised to about 220 mbar above atmospheric pressure and a high relative humidity (>90% RH) is established. When thermal equilibrium is re-established, the pressure in the chamber exhaust pipe is reduced to 65 mbar below atmospheric with a high-volume blower, and then the air inlet valve to the chamber is closed. A controlled adiabatic pressure reduction is then performed back down to 5 mbar overpressure, which progressively cools the air and forms a liquid or ice cloud when the RH rises above 100%. The pressure reduction is controlled with two regulation valves and two gate valves, which provide a selectable and highly flexible pressure profile lasting between 10 s and 10 minutes. The low exhaust pressure (-65 mbar) ensures sufficient pressure difference to drive the expansion down to a final 5 mbar chamber overpressure. Once the chamber reaches 5 mbar, the main air valve is re-opened to maintain the small over-pressure. During the expansion a cloud is maintained in the chamber and experiments are performed on cloud processing of aerosols, ice nucleation, and the effects of charge on cloud microphysics. The cycle can be repeated up to three or more times with a single CCN population, so the effects of multiple cloud processing of aerosol can be studied.

## 3    CLOUD thermal system

As well as precise control of precursor trace gases, UV intensity, ions, relative humidity and pressure, it is important to maintain good temperature uniformity and stability in the CLOUD chamber since many of the processes under study are sensitive to temperature, and contaminants can be released from the walls by positive temperature fluctuations. Temperature control is achieved by enclosing the chamber in a thermal housing through which air circulates at a precisely-controlled temperature between -70 °C and +100 °C. The 100 °C temperature is used for bakeout cleaning of the chamber walls between experimental campaigns. Figure 1 shows a schematic of the CLOUD thermal system and its components.

Here we present a study of the air temperature uniformity inside the CLOUD chamber during two campaigns. The first, in July 2014, was dedicated to calibration of the temperature sensors and evaluation of the thermal non-uniformities in the chamber under ideal conditions (no additional heat sources). The second was the CLOUD9 data campaign, September–November 2014. In addition, we calibrated a set of Pt100 calibration strings (PTH and PTV, see below) in the laboratory prior to the calibration campaign. Concerning the present paper, the two CLOUD campaigns are distinguished by the flow rate of fresh air and trace gases entering the chamber: 20 l/min and 210 l/min, respectively. CLOUD9 involved experiments to produce secondary organic aerosol particles, as well as expansions to form clouds. The high air plus trace gas inlet flow rate during the data campaign is required to compensate for the air continuously extracted by analysing instruments attached to the chamber. During both campaigns, the main air supply passes through a 10 m heat exchanger ("serpentine") pipe inside the CLOUD thermal housing (see below) to bring its temperature close to that of the chamber air before injection. However, during measurement campaigns, some trace gases are injected warm into the chamber at flow rates of around 1 l/min each, which influences the temperature uniformity inside the chamber. Trace gases are injected individually at the bottom of the CLOUD chamber, close to the main air inlet and just below the lower mixing fan. Only $O_3$ is mixed with the inlet air before entering the serpentine pipe heat exchanger to avoid localised high concentrations of $O_3$ in the chamber.

The temperature of the air inside the CLOUD chamber is measured by several precision thermometer strings: a) two horizontal strings oriented radially near the mid plane of the chamber, one using platinum resistance thermometers (Pt100, denoted PT), the other using thermocouple sensors (TC) and b) one vertical string that uses GaAs optical sensors (OS). The Pt100 sensors are four-wire sensors, with National Instruments (National Instruments Corp.) NI 9217 readout electronics. Figure 2 shows some details of the TC and OS strings. The thermocouple sensors are type K, with National Instruments (National Instruments Corp.) NI 9214 readout electronics. GaAs optical sensors (OTG-F with Pico-M single channel readout; Opsens Inc.) are used for the vertical temperature string since a high electrical field of up to 20kV/m in the chamber—to remove ions—rules out the use of conventional thermometer sensors with electrical wires. The thermocouple and optical sensors have a low mass (0.5 mm diameter sensor tip with 75 $\mu$m stainless steel wall thickness and 30 mm free length) and a fast response time: 3 s ($1/e$) in air. However, the Pt100 sensors have a higher mass (1.5 mm diameter stainless steel sheath with 100 $\mu$m wall thickness) and slower response time (180 s in air). Table 1 provides the positions in the CLOUD chamber of the temperature sensors of the five strings (TC/PTH, OS/PTV and PT).

Temperature differences over the large volume of the CLOUD chamber under steady run conditions are small and require precise ($\sim$0.01 °C) calibration of the temperature sensors. Removing the OS, TC and PT sensor strings for calibration would require disconnection from their readout electronics, which can result in a shift of their calibration. We therefore constructed dedicated horizontal

and vertical calibration strings (PTH and PTV, respectively) in which each Pt100 sensor had been
calibrated in close proximity to a certified WIKA (WIKA Alexander Wiegand SE & Co. KG) Pt100
reference thermometer, which itself was calibrated to $\pm 0.03$ °C absolute precision. The sensors
of PTH and PTV are four-wire Pt100 thermometers, with National Instruments NI 9217 readouts.
These sensors do not have the same high mass of the Pt100 sensors installed on the PT sensor string,
allowing for faster responses similar to the TC and OS sensors. After laboratory calibration, the
PTH and PTV calibration strings were then mounted in the CLOUD chamber alongside the TC and
OS strings, respectively, to transfer their calibrations *in situ*. The calibration strings, PTH and PTV,
were only installed during the temperature sensor calibration campaign, when no electric field, no
humidity and no trace gases were present. The calibration procedures (both in the laboratory and *in
situ*) are described in detail in Appendix A. All the measurements presented in the following analysis
correspond to calibrated sensor temperatures.

## 4  Temperature uniformity during calibration and NPF experiments

Figure 3 shows typical examples of the time series of thermometer sensors during NPF runs and
calibration periods. When the experimental conditions are not adjusted, the temperatures of individ-
ual sensors show drifts of only a few 0.01 °C over periods of several hours. Table 2 summarises
the experiments selected to characterise the chamber temperature uniformity during the CLOUD 9
campaign.

Figure 4 shows the temperature residuals of individual sensors from their mean values, after slow
trends in the data have been removed. The standard deviations of Gaussian fits to the data are 0.012
°C, 0.018 °C, and 0.004 °C for the TC, OS and PT strings, respectively. Comparison of the TC
and PT residuals shows that short term (<15 s) fluctuations of the air temperature in the CLOUD
chamber are very small (<0.01 °C). Furthermore, the comparably small OS residuals show that these
sensors are in principle capable of around 0.02 °C measurement precision. Supplemetary material
provides distribution of residuals for all other string temperature sensors.

Figure 5 shows representative temperature non-uniformities measured by several sensor strings in
the radial and vertical directions during calibration runs at 21 °C and 1 °C, respectively. For these
data, the temperature non-uniformity (maximum difference from the mean for the entire string)
measured by the Pt100 calibration strings is $\pm 0.01$ °C in the radial direction and $\pm 0.04$ °C in the
vertical direction. Comparison with the other strings shows close agreement of the TC string (panel
a) but somewhat larger residuals for the OS string (panel b), reflecting larger systematic errors in the
OS calibration.

Figure 6 shows the temperature non-uniformity in the horizontal and vertical directions at cham-
ber temperatures between -70 °C and 20 °C during calibration runs (filled circles and diamonds)
and steady conditions during data campaigns (hollow triangles). The temperature non-uniformity is

characterised as the maximum temperature difference of any sensor from the string mean, Max $\Delta$T.

A clear trend is seen in all temperature strings of increasing non-uniformity as the chamber temperatures is lowered, which results from an increased temperature difference between the chamber and the CERN experimental hall (around 20 °C). Nevertheless, under ideal (calibration) conditions the temperature uniformity is better than $\pm$0.06 °C in the radial direction and $\pm$0.1 °C in the vertical direction, for chamber temperatures between -70 °C and +20 °C. During experimental campaigns,

there is a high flow of fresh make-up air and trace gases (210 l/min) which can lead to higher non-uniformities in the vertical direction of up to $\pm$0.5 °C (green triangle symbols in panel b), depending on the temperature control of the make-up air. However, even during experimental campaigns, the radial temperature uniformity is better than $\pm$0.06 °C. Different fan speeds (up or down hollow triangles) do not show any influence on the temperature homogeneity.

**5  Temperature characteristics during cloud formation experiments**

Following an adiabatic pressure reduction, the temperature of the air in the chamber is below that of the walls. The wall temperature is essentially unaffected by the adiabatic cooling since its mass is several hundred times greater than the enclosed air mass. Therefore the walls transfer heat into the air and eventually bring it back into equilibrium at its original temperature, before the pressure

reduction took place. The rate of warming is a characteristic of the chamber surface area and volume, and determines how long a cloud can be maintained in the chamber before it evaporates. The rate of warming of the air can be described by a Newton's cooling law (Incropera and DeWitt, 2007):

$$\frac{\partial T}{\partial t} = -\frac{A\mu}{C}T = -\lambda T,$$

where $A$ is the area of the chamber, $\mu$ is the heat transfer coefficient between the walls and the

185 air, and $C$ is the heat capacity of the air. These constants are absorbed into a single reheating rate coefficient, $\lambda$, that characterises the CLOUD chamber. The reheating rate coefficient can be obtained by fitting the temperature versus time with an exponential curve given by

$$T(t) = T_w + (T_0 - T_w)e^{-\lambda t}, \tag{1}$$

where $T_w$ is the wall temperature and $T_0$ is the initial temperature at $t = 0$, immediately after the

190 adiabatic pressure reduction has finished. The reheating time constant $\tau = 1/\lambda$.

Around 300 adiabatic expansion experiments to form clouds were performed during CLOUD9. Figure 7 shows one example. The pressure reduction takes place over a period of 5 minutes, after which the temperature returns to its equilibrium value over the next 30 minutes. The red line shows an exponential fit to the reheating period (Eq. 1) with a time constant, $\tau = 200$ s. Figure 8 shows that

the air reheating rate is the same everywhere in the chamber and is the same for a wide range of experimental conditions. Figure 9 shows that $\lambda$ depends only weakly the initial temperature reduction, $\Delta T$.

During adiabatic expansions, air temperatures are up to 10 °C cooler than the walls, so large thermal non-uniformities may be anticipated. In Fig. 10 we show an example of the temperatures mea-
200 sured with vertical and horizontal strings during and after a fast (80 s) adiabatic pressure reduction at -30 °C. Compared with operation under equilibrium conditions (Fig. 6), larger non-uniformities of up to ±1.5 °C are present while the chamber returns to its equilibrium temperature. Turbulence during the expansion ensures efficient mixing of the chamber during the expansion, so the initial temperature non-uniformities are similar in the vertical and horizontal directions. Thermal non-uniformities
in the radial direction subsequently decrease as the air reheats and approaches the wall temperature. However, non-uniformities in the vertical direction reach a maximum around 2 minutes after the end of the expansion due to thermal stratification as the air reheats, with warmer air convecting to the upper part in the chamber (Fig. 10c). The radial string shows the effect of relatively warm make-up air entering the chamber after the expansion has finished and the main air inlet valve has been re-
opened to maintain the baseline chamber pressure at +5 mbar. This can be seen in Fig. 6d by the higher temperatures of TC5 and TC6, which are closest to the axis of the chamber and mixing fans. The flow of relatively warm make-up air partly contributes to the vertical stratification since the air exhaust pipes are located at the top of the chamber.

## 6 Conclusions

In order to characterise the temperature uniformity of the air inside the CERN CLOUD chamber, we have constructed and calibrated several thermometer strings using various sensors (Pt100, thermocouple and optical/GaAs). Our measurements show that, under stable calibration conditions, the temperature uniformity is better than ±0.06 °C in the radial direction and ±0.1 °C in the vertical direction, for chamber temperatures between -70 °C and +20 °C. This excellent performance for a
large-volume (26.1 m$^3$) chamber underscores the quality of the CLOUD thermal control system and thermal housing. Moreover, during periods when the experimental conditions are not adjusted, the chamber air drifts by only a few 0.01 °C. During experiments, there is a high flow of fresh make-up air and trace gases—up to around 210 l/min—to compensate for the air extracted into sampling instruments. This can lead to higher thermal non-uniformities unless the make-up air is carefully ad-
justed to match the chamber temperature before injection. Larger non-uniformities of up to ±1.5 °C occur during adiabatic expansions to form clouds in the chamber, since the walls are up to 10 °C warmer than the enclosed air. After an adiabatic expansion, the chamber air is reheated by the walls and returns to its equilibrium temperature with a time constant of around 200 s.

## Appendix A: Thermometer sensor calibrations

**Calibration of the horizontal and vertical Pt100 calibration strings**

Careful calibration of all thermometer sensors used in the CLOUD chamber is required to extract meaningful results from the thermal measurements. Two calibration Pt100 strings were specially constructed to allow in-situ calibration of the permanent CLOUD temperature sensors (TC, OS and PT). One vertical string (PTV) was used to calibrate the vertical OS string and one horizontal string

(PTH) to calibrate the horizontal TC and PT strings. Pt100 sensors were used in the PTH and PTV strings due to their precision, reliability, temperature range and well-defined calibration procedure. Calibration of the sensors on the PTV and PTH calibration strings (PTH and PTV, respectively) was carried in the laboratory in July 2014. After assembly and calibration of the PTH and PTV strings, they were mounted in the chamber alongside the TC, PT and OS strings, and then used to calibrate

the latter strings *in situ*, as described below.

The calibration strings are designed to allow the sensors to be detached from the string without disconnection from their readout electronics (Fig. A1). The wiring from the sensors to the readout electronics passes through the support tube for the string. The cables have sufficient length to allow the sensors to be detached and brought together. In this way the sensors can be placed in close

proximity with a reference Pt100 during inter-calibration in water baths or liquid nitrogen, without disconnection from their readout electronics. A certified WIKA (WIKA Alexander Wiegand SE & Co. KG) Pt100 reference thermometer—calibrated according to ISO standard IEC751—was used as the absolute reference to calibrate the individual PTH and PTV sensors. The WIKA Pt100 reference thermometer is calibrated to 0.03 °C absolute temperature uncertainty in the range 0–100 °C.

The PTH and PTV sensors were calibrated according to the Calendar-Van Dusen (CVD) equation Callendar (1887); Dusen (1925), which relates the resistance, $R$ ($\Omega$) and temperature, $T$ ( °C) of a platinum resistance thermometer by:

$$R(T) = \begin{cases} R_0\left[1 + A\,T + B\,T^2\right] & \text{if } T \geq 0 \\ R_0\left[1 + A\,T + B\,T^2 + (T - 100)\,C\,T^3\right] & \text{if } T < 0 \end{cases} \tag{A1}$$

The standard values for a Pt100 sensor are as follows: $R_0 = 100\ \Omega$, $A = 3.908 \times 10^{-3}\ \Omega\,°C^{-1}$, $B =$

$-5.775 \times 10^{-7}\ \Omega\,°C^{-2}$ and $C = -4.183 \times 10^{-12}\ \Omega\,°C^{-3}$ Commission et al. (2008). We determined fitted values of these parameters for each sensor in the calibration procedure, as described below.

All calibration string sensors were wired through their respective strings and connected to their readout electronics. The sensors were then placed in close proximity with the reference WIKA thermometer in a water bath, ensuring no contact was made between sensors. The water bath was filled

with ultrapure water from the CLOUD humification system. CLOUD's cleanliness standards do not allow the use of organic solvents or brine as thermostat liquid, and so only pure water and liquid nitrogen were available for the laboratory calibration. A Monte Carlo analysis of the calibration was

made in order to ensure the calibration provided an acceptable measurement uncertainty at temperatures inside the whole range of measurements made in CLOUD (see below).

The Huber CC-K15 liquid bath (Huber Kältemaschinenbau AG) controlled the water temperature to better than 0.02 °C stability over the calibration range from 2 °C to 70 °C. A 0 °C point was obtained using a mixture of CLOUD ultrapure water and ice. The WIKA reference thermometer confirmed that the temperature was 0 °C. For calibration at lower temperatures, the sensors and reference WIKA sensor were placed in a dewar flask filled with liquid nitrogen. An electrical barometer

was used to measure the atmospheric pressure. The temperature measured at the boiling point of liquid nitrogen agreed within 0.01 °C with standard value for nitrogen at the measured atmospheric pressure (-196.21±0.0096 °C at 966±1 mbar). For the calibration we used the standard value of the liquid nitrogen boiling point at the measured atmospheric pressure.

A least squares fit of Eq. A1 was then applied to the calibrated temperature measurements for

each sensor. The $C$ parameter was fixed at the standard value since it was poorly constrained by the calibration points. Moreover, propagation of the uncertainties showed that uncertainties in $B$ have a larger effect than uncertainties in $C$. The residuals of the fit are shown in Fig. A2, and the fitted parameters and errors and uncertainties are summarised in Table A1.

We used a Monte Carlo method to verify that, after applying the CVD fits, the uncertainties in the

280 calibration string sensors were negligible. From table A1, each sensor's estimated parameters were used to create a lookup table between resistance (R) and temperature (T) for the range -70 °C to 70 °C (a range spanning all of the current CLOUD measurements), using equation A1. The same estimated parameters were then assumed to follow a normal distribution of values around its fitted value, with a standard deviation equal to its uncertainty. From those distributions a set of 1000 trios of $R_0^{fit}$,

$A^{fit}$ and $B^{fit}$ were retrieved. Using each trio, a similar lookup table was created using A1. Using both lookup tables, the temperature at the same resistance value was compared for all 1000 trios. The average temperature difference represents the uncertainty introduced in the measurements by fitting the Calendar-Van Dusen coefficients. These differences are plotted in figure A3 and show that, after applying the CVD fits, almost all calibration sensors have an uncertainty below 0.01 °C over the full

CLOUD temperature range and so are well-matched for *in-situ* calibration of the CLOUD chamber strings.

**Calibration of the thermocouple, optical sensor and horizontal PT100 string**

After calibrating the calibration sensors (PTV and PTH) in the laboratory, the calibration strings were mounted inside the CLOUD chamber. Each calibration string was built so that each calibration

sensor would occupy the vertical and radial position as close as possible to the position of the sensor to be calibrated. In this way, each TC and OS sensor now had a calibrated sensor in close proximity to define the calibrated temperature at that location. A total of 10 experiments were performed at

temperatures ranging from -60 °C to 70 °C. Each experiment consisted of setting a desired chamber temperature and acquiring data for around 24 hours.

To calibrate the TC and OS sensors, the difference between the temperature measured by each sensor $(T_s)$ and its adjacent calibrated Pt100 sensor $(T^*)$ was fitted to a polynomial:

$$T^* - T_s = \sum_{n=0}^{k_s} x_n T_s^n, \tag{A2}$$

where $x_n$ are fitted coefficients and $k_s$ is the degree of the polynomial fit required for each sensor. The fitted parameters were subsequently used to correct each measured sensor temperature to its calibrated value.

A least squares fit of Eq. A2 was applied to the calibrated temperature measurements for each TC sensor. A second order fit was found to best describe the data (Fig. A4a). The TC fit residuals are shown in Fig. A4b. We summarise the fitted calibration parameters and uncertainties for the thermocouple string in Table A2. PTH2 was non responsive after being placed in the chamber. Thus its measurement was replaced by a spatial linear fit between the measurements of PTH1 and PTH3 defined by:

$$T^*_{PTH2} = T_{PTH1} + (r_{PTH2} - r_{PTH1}) \frac{T_{PTH3} - T_{PTH1}}{r_{PTH3} - r_{PTH1}} \tag{A3}$$

where $r_{PTHi}$ and $T_{PTHi}$ are, respectively, the radial position and temperature measured by the PTH sensor of index $i$, as defined in Table 1.

The optical sensors were calibrated in a similar way. Here a 3rd order polynomial was used, except for OS4 and OS5, which required a 4th order polynomial (Fig. A5a). OS4 displayed anomalous behavior, and to a lesser extent, also OS5. Both were calibrated but ignored for analysis of the temperature uniformity in the CLOUD chamber. The OS fit residuals are shown in Fig. A5b and are somewhat larger than those obtained for the TC sensors. We summarise the fitted calibration parameters and uncertainties for the optical sensor string in Table A3.

The PT sensors were calibrated using a least squares fit to a linear function of the measured temperature $(T_{PT})$:

$$T(T_{PT}) = x_1 T_{PT} + x_0 \tag{A4}$$

Since the PT string sensor positions did not contain a set of calibration sensors in their vicinity, a spatial linear interpolation of the values of the horizontal calibration string was made similar to equation A3 in order to create a virtual calibration string at the positions of the sensors in the PT string. The PT string contained originally 5 sensors, but PT4 was damaged prior to this study. The results of these fits are summarised in Table A4.

*Acknowledgements.* This research has been supported by a Marie Curie Initial Training Network Fellowship of the European Community's Seventh Framework Programme under contract number (PITN-GA-2012-316662-CLOUD-TRAIN.)

330

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

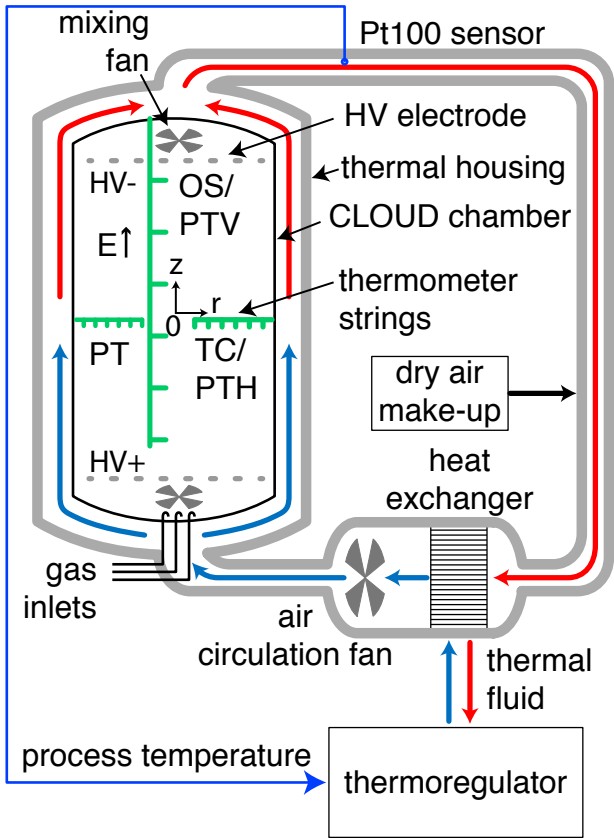

**Figure 1.** Schematic of the CLOUD thermal system. Thermally-controlled air is circulated by a fan at a rate of around 7200 $m^3$/h within a thermally-insulated housing surrounding the stainless steel CLOUD chamber (3 m diameter, 26.1 $m^3$ volume). The air is returned to the bottom of the chamber along an insulated duct, where it passes through a heat exchanger. A small flow of desiccated air is added to compensate for leaks and ensure that the dew point of the re-circulated air remains well below -80 °C. The air is precisely maintained at a selectable "process temperature" in the range between -70 °C and +100 °C by a 36 kW thermoregulator (Unistat 850W; Huber Kältemaschinenbau AG) which controls the temperature of thermal fluid circulating through the heat exchanger. Fresh make-up air and trace gases continuously enter the chamber to compensate for sampling losses and maintain 5 mbar over-pressure inside the chamber. The chamber air is injected underneath the lower mixing fan (z=0 m) at rates between 20 l/min (during sensor calibration) and 210 l/min (during CLOUD campaigns). During data campaigns, vertical electrical fields of up 20 kV/m are generated inside the CLOUD chamber by a pair of HV electrodes whose mid-plane (at 0 V potential) coincides with the mid plane of the chamber. The temperature of the air inside the CLOUD chamber is measured by several precision thermometer strings, indicated by green lines. The $rz$ cylindrical coordinate axes are indicated in the figure, with the origin in the centre of the chamber, as a guide for the sensor positions indicated in Table 1

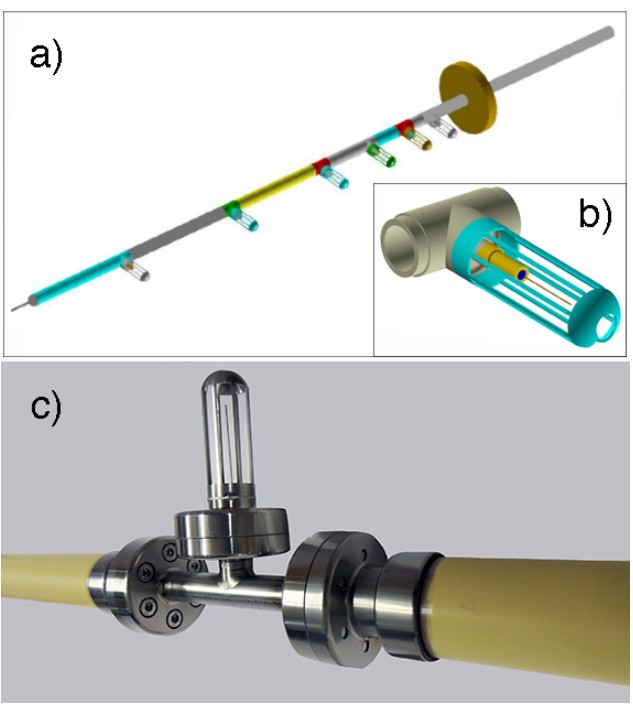

**Figure 2.** Images of two temperature strings showing a) a model drawing of the thermocouple (TC) string with 6 TC sensors and one Pt100 sensor at the tip, mounted on a 1.25 m stainless steel support tube welded to a DN100 flange, b) a model drawing of a single TC sensor (thin brown line), and c) a photograph of an approximately 25 cm section of the optical sensor string showing a sensor mounted inside a stainless steel capillary and transition structure that is welded to two partially-conducting zirconia ceramic tube spacers of 27 mm outer diameter (yellow). The sensors are mounted inside Faraday cages (e.g. coloured blue in panel b) to protect against corona discharge when a 20 kV/cm vertical electric field is present in the CLOUD chamber. Ultra-clean materials (stainless steel and ceramics) are used throughout, with the readout wires/optical fibers passing inside the hollow support structures.

**Table 1.** Position of the temperature sensors in CLOUD. The CLOUD chamber is a cylinder of 150 cm radius ($r$) and approximately 400 cm height ($z$). The origin of the cylindrical coordinate system is located in the centre of the chamber, with the $z$ axis pointing vertically upwards (Fig. 1). The calibration strings, PTH and PTV, have Pt100 sensor locations that are located within about 1 cm of the indicated sensor locations for the TC and OS strings, respectively. The calibration strings, which are only installed during calibration measurements, are displaced laterally from the TC string by $\Delta z$ = -20 cm (PTH) and from the OS string by 10 cm azimuthally (PTV).

| | Horizontal strings: Thermocouple (TC) and Calibration (PTH; ±1 cm) | | Vertical strings: Optical (OS) and Calibration (PTV; ±1 cm) | | Horizontal string: Pt100 (PT) | |
|---|---|---|---|---|---|---|
| Sensor # | Radius, $r$ (cm) | Height, $z$ (cm) | Radius, $r$ (cm) | Height, $z$ (cm) | Radius, $r$ (cm) | Height, $z$ (cm) |
| 1 | 145 | 0 | 50 | 123.1 | 145 | 0 |
| 2 | 138 | 0 | 50 | 78.3 | 115 | 0 |
| 3 | 128 | 0 | 50 | 33.5 | 90 | 0 |
| 4 | 115 | 0 | 50 | -11.3 | - | - |
| 5 | 90 | 0 | 50 | -56.1 | 60 | 0 |
| 6 | 60 | 0 | 50 | -100.9 | - | - |

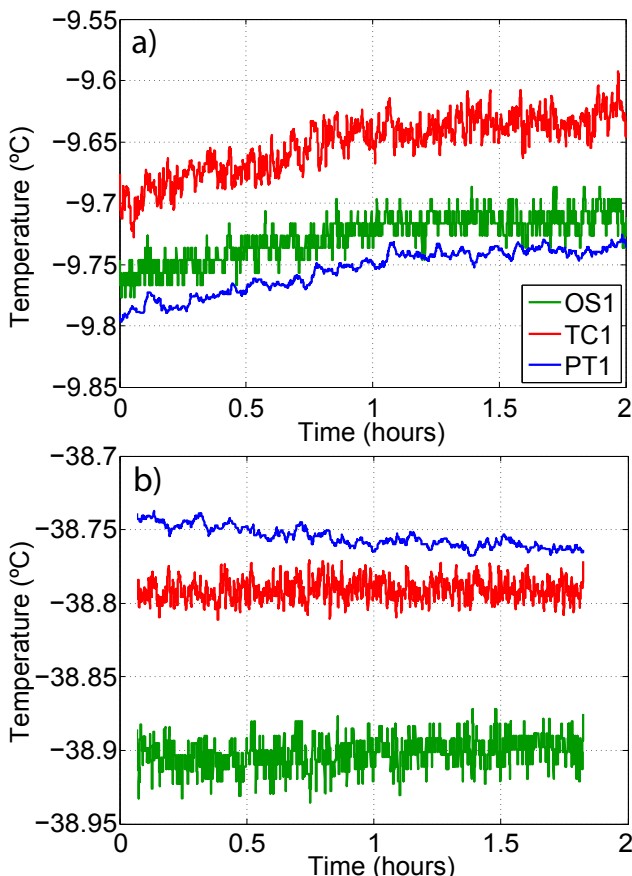

**Figure 3.** Examples of the typical temperature stability in the CLOUD chamber over a period of several hours: a) during an experimental data run and b) during a calibration run (OS, optical sensor; TC, thermocouple, TC; and PT, Pt100 sensor). The temperature drift in panel a) can result from setting new experimental conditions such as the addition of ultra violet radiation or trace gases, which causes a slight change in equilibrium temperature. The measurements are smoothed with a 5 s fixed median window to reduce point-to-point scatter.

**Table 2.** Experiments selected to characterise the chamber temperature uniformity during the CLOUD9 campaign. The experiments were selected to have 3-hour periods at various temperatures during which no changes of the gas or other conditions were made.

| Date of experiment | Temperature (°C) | Fan speed (%) | Duration (min) |
|---|---|---|---|
| 29 September 2014 | -10 | 21 | 185 |
| 8 October 2014 | -30 | 21 | 180 |
| 25 October 2014 | -10 | 100 | 180 |
| 26 October 2014 | -20 | 100 | 180 |
| 27 October 2014 | -20 | 21 | 180 |
| 29 October 2014 | -40 | 100 | 180 |
| 29 October 2014 | -40 | 21 | 180 |

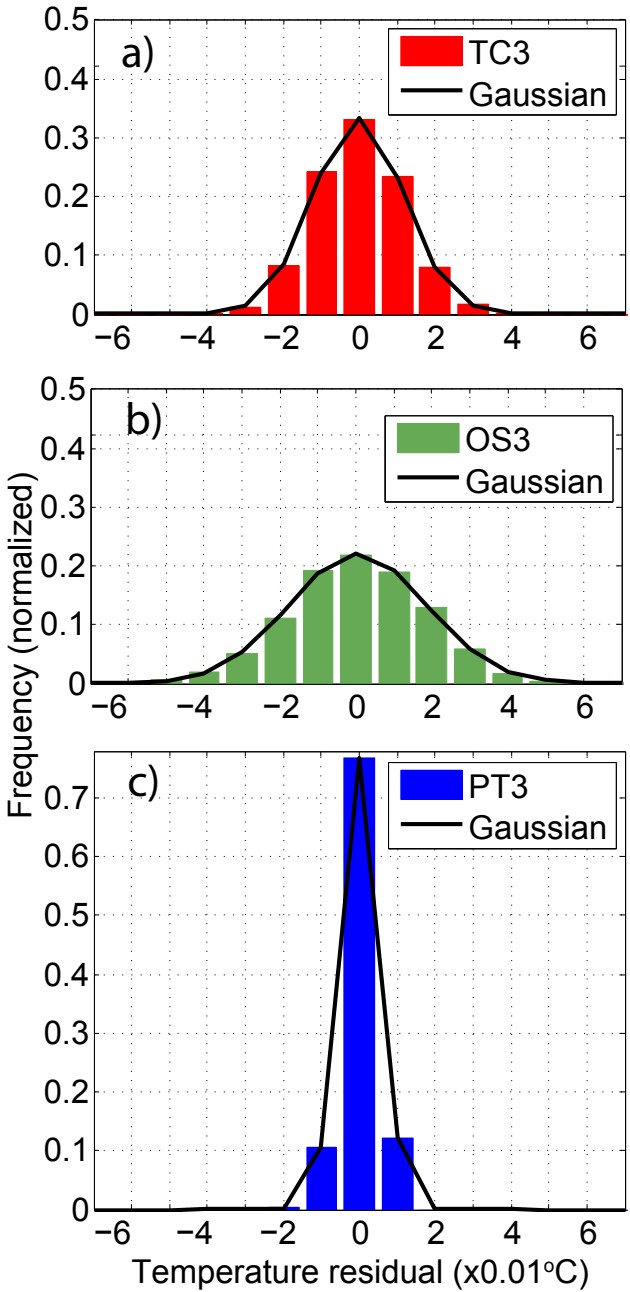

**Figure 4.** Examples of the characteristic temperature residuals from their mean values over periods of several hours during calibration runs for a) TC3 at -40 °C, b) OS3 at -0 °C and c) PT3 at -30 °C. Slow trends in the data have been removed, and the measurements are smoothed with a 5 s fixed median window. The standard deviations of Gaussian fits to the data are a) 0.012 °C, b) 0.018 °C, and c) 0.004 °C. See supplement material for distributions of other sensors.

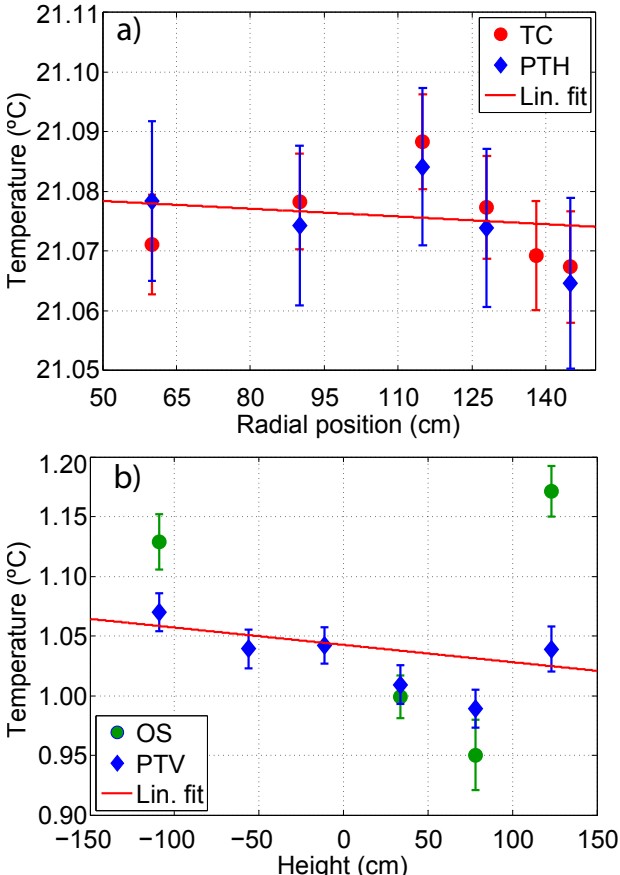

**Figure 5.** Comparison of the temperature non-uniformity measured by several sensor strings in a) the radial and b) the vertical directions during calibration runs at 21 °C and 1 °C, respectively. Linear fits are shown to guide the eye. OS4 and OS5 are not included in panel b (see Appendix A for details). The error bars in this and other figures show 1 sigma statistical uncertainties and do not account for possible systematic uncertainties.

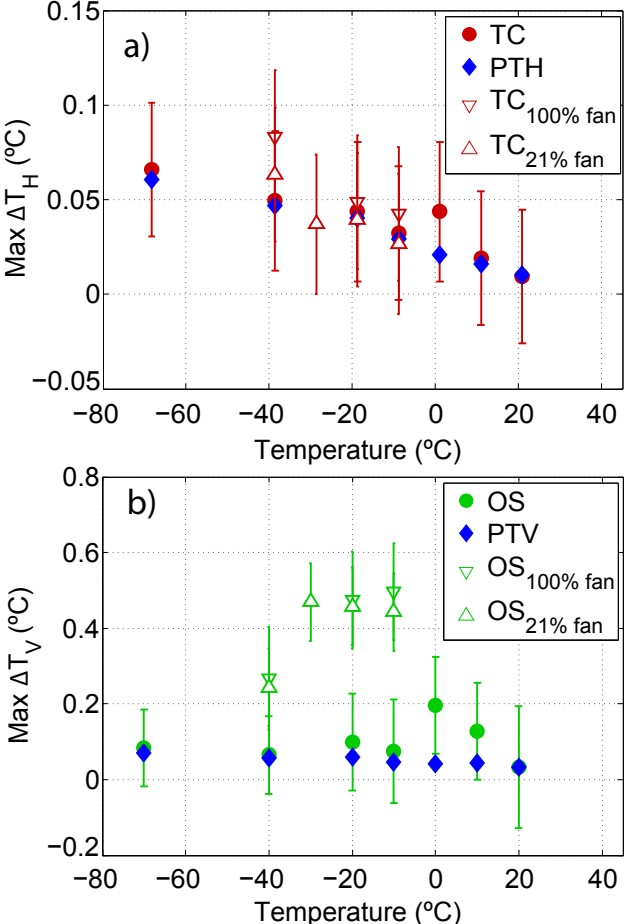

**Figure 6.** Maximum temperature difference from the string mean (temperature non-uniformity), $\Delta T$, in the a) horizontal and b) vertical directions at chamber temperatures between -70 °C and 20 °C. The data were recorded during both calibration (solid symbols) and data-taking runs (hollow symbols) in June and September–October 2014, respectively.

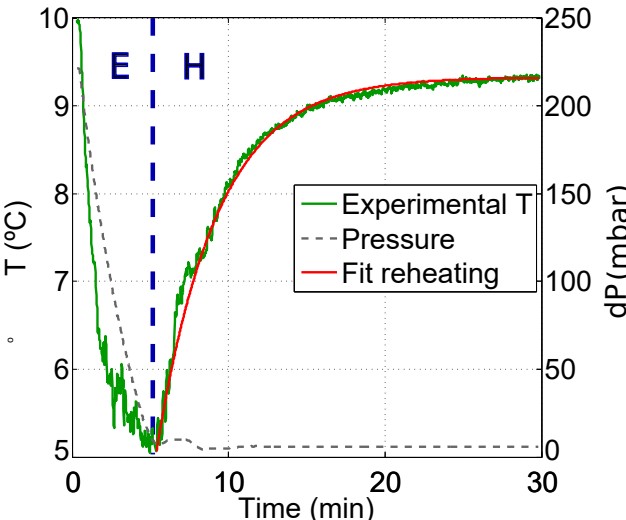

**Figure 7.** Example of an adiabatic pressure reduction to form a cloud in the CLOUD chamber. The chamber pressure is reduced from 220 mbar to 5 mbar above one atmosphere pressure during a period of 5 minutes (dashed curve and right-hand scale). This produces of reduction of the air temperature by around 5 °C (green curve and left-hand scale, recorded by a TC sensor). Provided the initial relative humidity is sufficiently high, a liquid cloud forms in the chamber during the cool period. The air in the chamber then returns to its equilibrium temperature set by the relatively warm chamber walls, and the cloud eventually evaporates. In this example the initial air temperature, before expansion, had not yet reached the equilibrium value near 9.3 °C. The red line is an exponential fit to the warming period with a time constant of 200 s (Eq. 1).

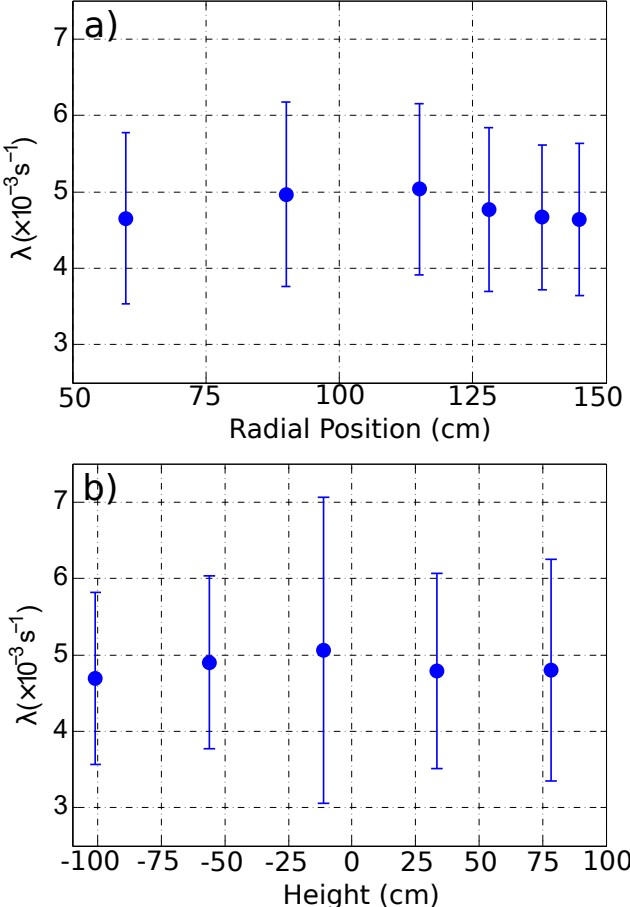

**Figure 8.** Air reheating rates ($\lambda$) following an adiabatic pressure reduction versus a) radial and b) vertical position in the chamber. The data points show the mean and standard deviations of the air reheating rates obtained for 300 expansions in the CLOUD9 campaign.

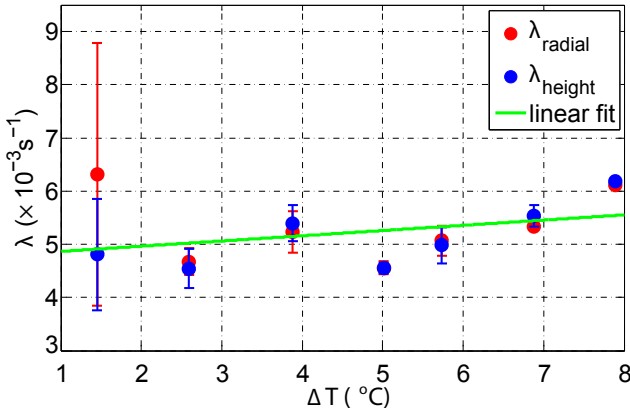

**Figure 9.** Air reheating rates ($\lambda$) following an adiabatic pressure reduction versus the initial temperature reduction.

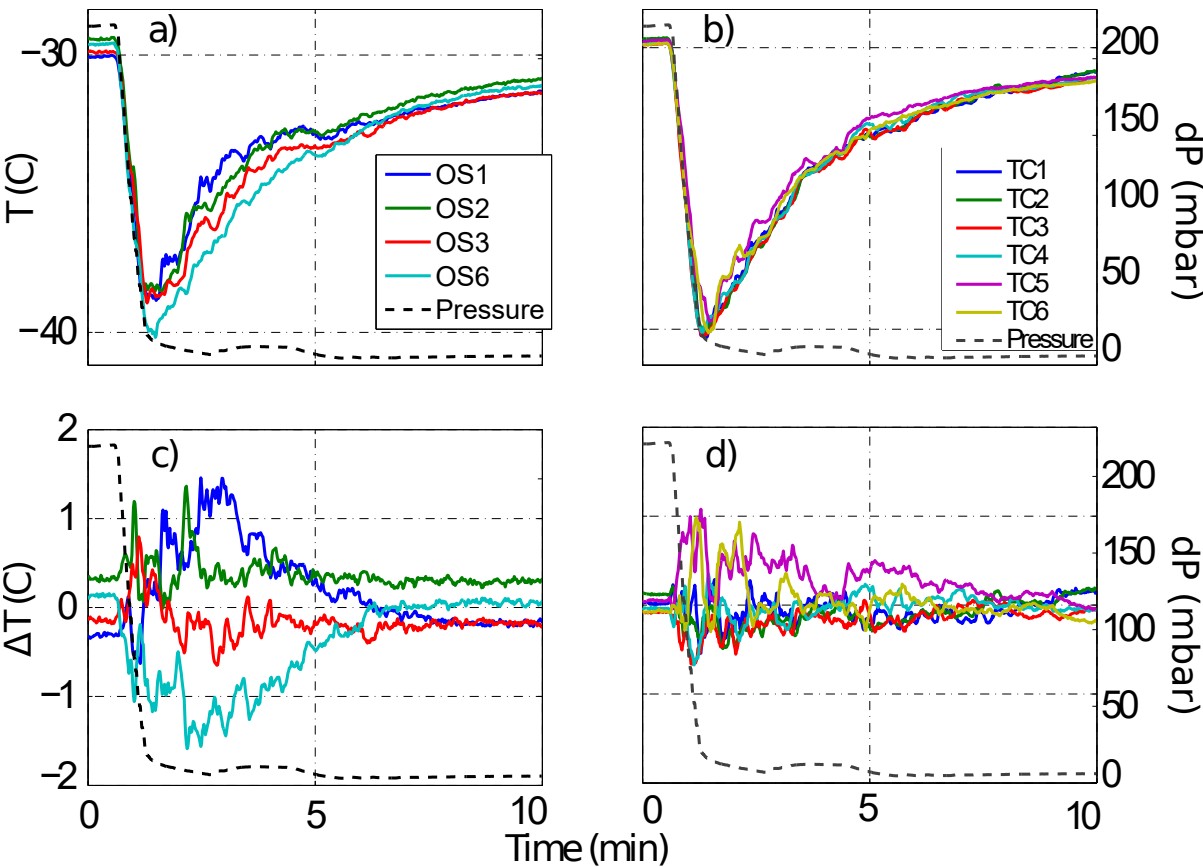

**Figure 10.** Temperature non-uniformities following an adiabatic pressure reduction during CLOUD9 at -30 °C in a) the vertical direction (OS) and b) the horizontal direction (TC). Panels c and d show the residuals from the mean temperatures in the vertical and horizontal directions, respectively. The relative air pressures are shown by dashed curves. Compared with operation under equilibrium conditions (Fig. 6), larger non-uniformities of up to ±1.5 °C in the vertical direction and ±1 °C in the horizontal direction are present while the chamber returns to its equilibrium temperature with time constant of about 200 s.

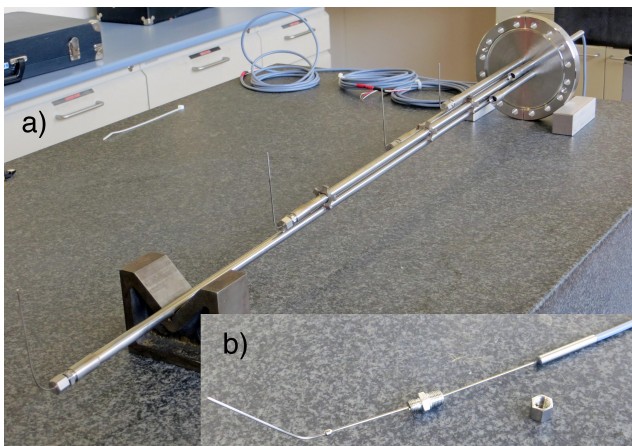

**Figure A1.** The horizontal calibration string (PTH) showing a) the string during assembly, with 4 of the 6 Pt100 sensors mounted and b) a single Pt100 sensor before mounting in the string (a 1.5 mm diameter stainless steel sheath surrounds the actual Pt100 sensor). The calibration strings are designed to allow the sensors to be dismounted without disconnection from their readout electronics.

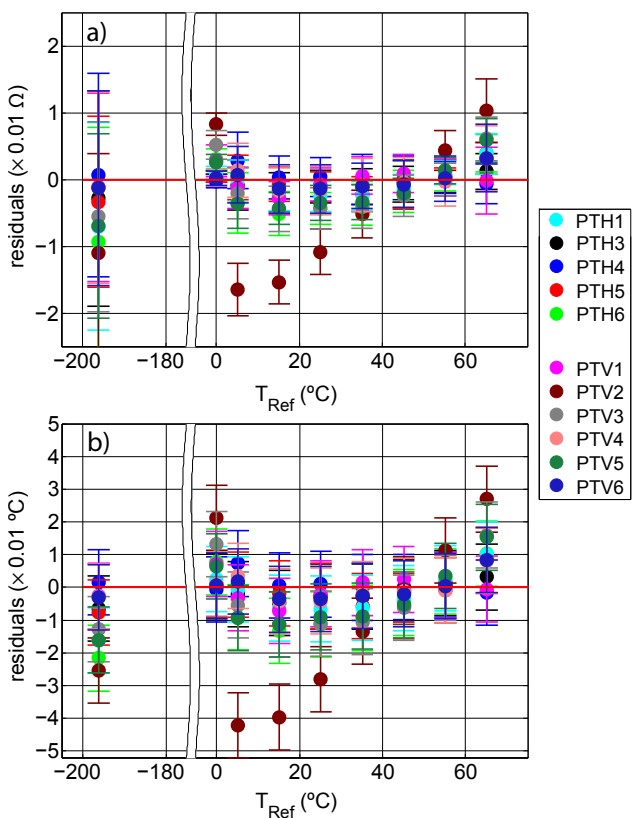

**Figure A2.** Residuals of the least squares fits to the CVD equation (Eq. A1 and Table A1), for the horizontal (PTH) and vertical (PTV) calibration strings versus temperature: a) resistance residuals and b) temperature residuals. The data were obtained during the laboratory calibration, July 2014.

**Table A1.** Fitted parameters and one standard deviation uncertainties of the Callendar-van-Dusen (CVD) coefficients (Eq. A1) for the horizontal and vertical calibration strings, PTH and PTV, obtained during the laboratory calibration, July 2014. The parameter $C$ was fixed at $-4.183 \times 10^{-12}$ $\Omega$ $°C^{-3}$.

|  | $R_0^{fit}$ | $\sigma R_0$ | $A^{fit}$ | $\sigma A$ | $B^{fit}$ | $\sigma B$ | Fit $R^2$ |
|---|---|---|---|---|---|---|---|
| PTH1 | 100.0322 | 9.36E-4 | 3.908E-3 | 3.25E-07 | -6.62E-07 | 3.70E-09 | 0.9991 |
| PTH3 | 100.0216 | 2.55E-4 | 3.908E-3 | 1.27E-07 | -6.70E-07 | 1.55E-09 | 0.9998 |
| PTH4 | 99.98723 | 3.06E-4 | 3.914E-3 | 1.03E-07 | -6.54E-07 | 1.06E-09 | 0.9999 |
| PTH5 | 100.0453 | 6.86E-4 | 3.907E-3 | 2.26E-07 | -6.60E-07 | 2.35E-09 | 0.9995 |
| PTH6 | 100.0182 | 1.98E-3 | 3.913E-3 | 5.80E-07 | -6.46E-07 | 5.80E-09 | 0.9982 |
| PTV1 | 100.0496 | 1.11E-3 | 3.910E-3 | 3.17E-07 | -6.61E-07 | 2.52E-09 | 0.9996 |
| PTV2 | 100.0364 | 5.26E-3 | 3.908E-3 | 1.59E-06 | -6.87E-07 | 1.35E-08 | 0.9894 |
| PTV3 | 100.0503 | 2.61E-3 | 3.909E-3 | 6.64E-07 | -6.69E-07 | 5.46E-09 | 0.9981 |
| PTV4 | 100.0355 | 5.17E-4 | 3.912E-3 | 1.81E-07 | -6.49E-07 | 1.52E-09 | 0.9999 |
| PTV5 | 100.0184 | 1.79E-3 | 3.910E-3 | 5.32E-07 | -6.80E-07 | 5.19E-09 | 0.9979 |
| PTV6 | 100.0108 | 3.91E-4 | 3.912E-3 | 1.51E-07 | -6.48E-07 | 1.37E-09 | 0.9999 |

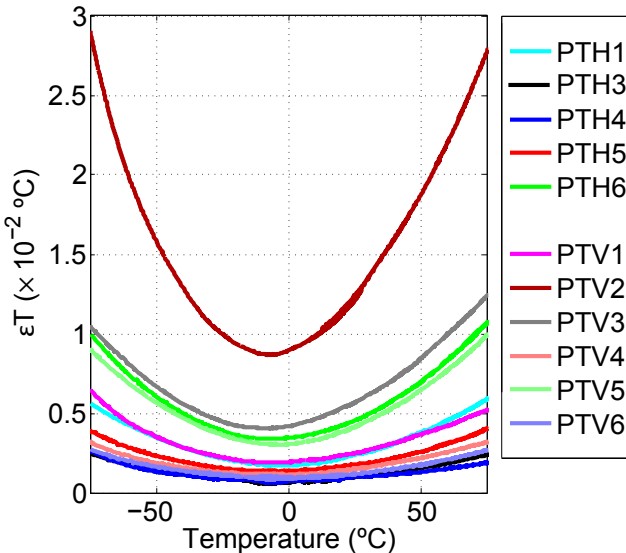

**Figure A3.** Estimated uncertainty of the temperature measurement of each calibration sensor versus temperature, based on the individual fits of the laboratory calibration measurements to the CVD equation. After calibration, almost all sensors have an uncertainty below 0.01 °C over the indicated temperature range.

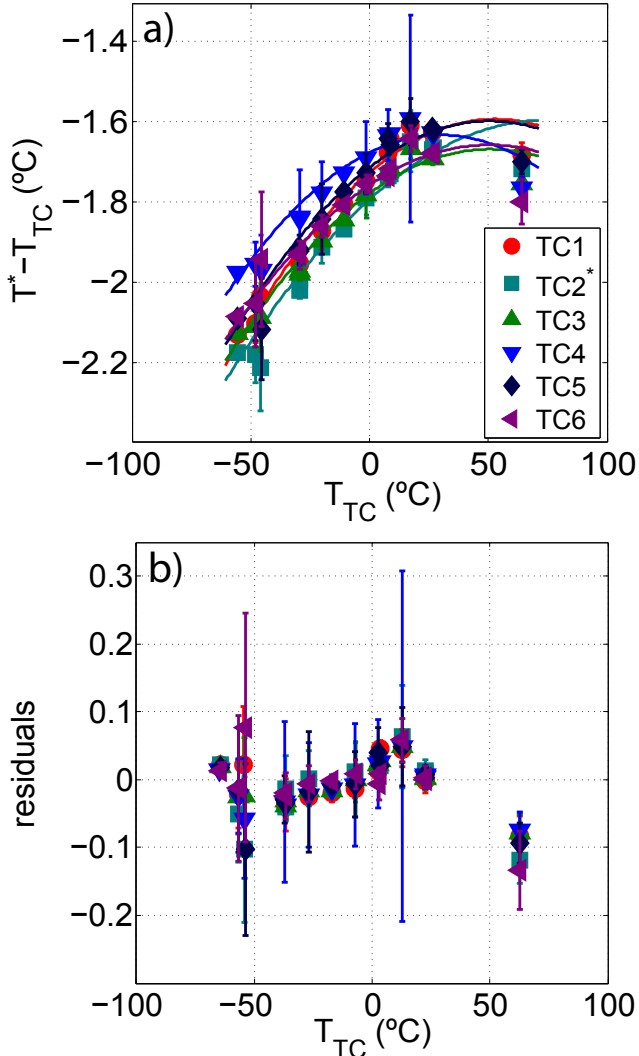

**Figure A4.** Calibration of the TC string during during the CLOUD calibration campaign: a) fitted calibration curves (Eq. A2 with $k_s = 2$) and b) residuals from the calibration fit. There was a malfunction of the Pt100 sensor (PTH2) for TC2 so the PTH2 values were obtained by interpolation between PTH1 and PTH3.

**Table A2.** Fitted calibration parameters and one standard deviation uncertainties for the thermocouple string obtained during the CLOUD calibration campaign.

| Sensor | $x_2$ | $\sigma x_2$ | $x_1$ | $\sigma x_1$ | $x_0$ | $\sigma x_0$ | $\chi^2/\nu$ |
|---|---|---|---|---|---|---|---|
| TC1 | -6.39E-5 | 1.28E-5 | 4.47E-2 | 4.40E-04 | -1.70 | 1.50E-02 | 0.34 |
| TC2 | -6.04E-5 | 1.36E-5 | 4.27E-2 | 7.74E-04 | -1.74 | 1.70E-02 | 0.44 |
| TC3 | -6.17E-5 | 1.01E-5 | 3.85E-2 | 3.46E-04 | -1.75 | 1.20E-02 | 0.25 |
| TC4 | -6.39E-5 | 1.56E-5 | 2.35E-2 | 1.10E-03 | -1.66 | 1.80E-02 | 0.71 |
| TC5 | -6.55E-5 | 2.17E-5 | 4.01E-2 | 7.23E-04 | -1.68 | 2.50E-02 | 1.41 |
| TC6 | -6.51E-5 | 1.31E-5 | 2.93E-2 | 4.45E-04 | -1.72 | 1.60E-02 | 0.46 |

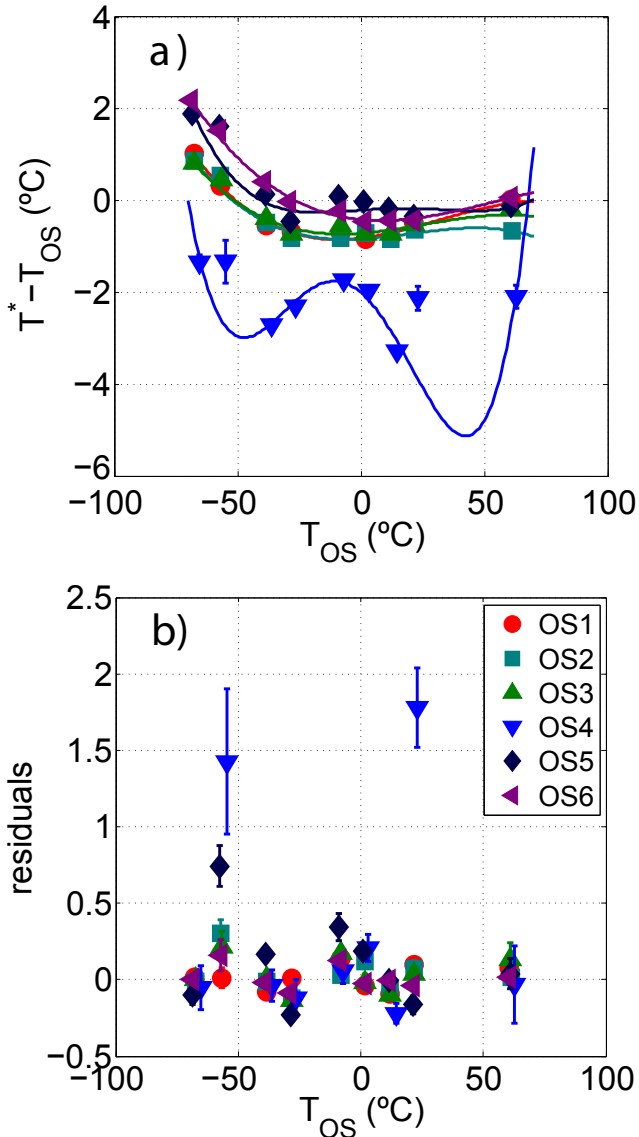

**Figure A5.** Calibration of the OS string during during the CLOUD calibration campaign: a) fitted calibration curves (Eq. A2 with $k_s = 3$ or $4$) and b) residuals from the calibration fit.

**Table A3.** Fitted calibration parameters and one standard deviation uncertainties for the optical sensor string obtained during the calibration campaign.

| Sensor | $x_4$ | $\sigma x_4$ | $x_3$ | $\sigma x_3$ | $x_2$ | $\sigma x_2$ | $x_1$ | $\sigma x_1$ | $x_0$ | $\sigma x_0$ | $\chi^2/\nu$ |
|---|---|---|---|---|---|---|---|---|---|---|---|
| OS1 | N/A | N/A | -3.81E-6 | 2.93E-6 | 2.68E-4 | 1.48E-4 | 9.60E-3 | 9.30E-3 | -7.42E-1 | 2.01E-1 | 28.40 |
| OS2 | N/A | N/A | -4.35E-6 | 1.97E-6 | 1.75E-4 | 8.66E-5 | 7.34E-3 | 6.60E-3 | -7.65E-1 | 1.33E-1 | 7.02 |
| OS3 | N/A | N/A | -2.89E-6 | 2.23E-6 | 2.30E-4 | 1.04E-4 | 5.74E-3 | 7.10E-3 | -7.22E-1 | 1.86E-1 | 12.34 |
| OS4 | 2.54E-7 | 2.52E-7 | -1.01E-6 | 8.71E-6 | -1.04E-3 | 1.00E-3 | 1.12E-3 | 2.76E-2 | -1.74 | 0.46 | 4.67 |
| OS5 | 1.36E-7 | 3.48E-7 | -1.09E-6 | 1.23E-5 | 3.77E-4 | 1.50E-3 | 5.27E-3 | 3.89E-2 | -3.25E-2 | 6.97E-1 | 45.72 |
| OS6 | N/A | N/A | -2.16E-6 | 2.66E-6 | 3.27E-4 | 1.27E-4 | 4.64E-3 | 7.50E-3 | -408E-1 | 1.74E-1 | 19.32 |

**Table A4.** Fitted calibration parameters and one standard deviation uncertainties for the Pt100 string during the calibration campaign.

|  | $x_0$ | $\sigma x_0$ | $x_1$ | $\sigma x_1$ | Fit $R^2$ |
|---|---|---|---|---|---|
| PT1 | 0.561 | 0.020 | 1.011 | 0.001 | 1.00 |
| PT2 | 0.187 | 0.083 | 1.012 | 0.002 | 1.00 |
| PT3 | 0.453 | 0.020 | 1.007 | 0.001 | 1.00 |
| PT5 | 0.335 | 0.025 | 0.998 | 0.001 | 1.00 |