# Peer review of "Temperature uniformity in the CERN CLOUD chamber"

_Atmospheric Measurement Techniques, 2017_

## Referee Comment (RC1) · Anonymous Referee #1 · 2 Jul 2017

Review of
Dias et al.:
**Temperature uniformity in the CERN CLOUD chamber**

This manuscript describes the temperature calibration and measurements inside the CLOUD chamber at CERN. Using different temperature sensor arrays in the horizontal and vertical the uniformity of temperature inside the chamber is characterised.

I highly appreciate the effort to perform this kind of chamber characterisation, as the authors correctly state; temperature is a very important factor in cloud nucleation. Thus, I find this kind of manuscript very important and well suited for publication in AMT, however, this manuscript needs a thorough overhaul, as I will explain in the following.

**Major questions:**

After reading the manuscript several times, now have the impression that during the calibration campaign only runs with steady flow conditions have been performed. Is this true? If so, why did you not perform evacuations for cloud formation as well? How much data (e.g. in hours, or if evacuations how many) did you take during the calibration campaign. How many hours did you measure in flow conditions during the data campaign? I think this information would be useful and should be added to the manuscript. Please clarify! In general, more information about the experiments performed during both campaigns might be helpful, maybe summarise the experiments in a table?

Why Pt100 sensors were chosen for the calibration strings? This is not motivated in the manuscript. However, as you state, they have a rather long response time (180 s), which seems insufficient for cloud experiments, where temperature drops much faster than this? How do you compare temperatures from the fast response sensors to these slow response sensors? Please explain!

Why do you average the data over 15sec? Does it make sense in case of the Pt100 with time constants around 180sec? On the other hand, 15secs smooth out fluctuations in the fast sensors responses. Do you also smooth data during evacuations?

I would like to see a schematic of CLOUD chamber that shows more detail, e.g. the "serpentine" pipe (page 3, line 71), regulation and gate valves (line 51), sampling ports for cloud measurements. Particularly the valve positions in respect to the temperature sensors would be good to know!

A large part of the introduction (i.e. from line 28) reads more like a potential chapter 2 "chamber operation" (or similar). I would expect more introduction about cloud chambers and the importance of temperature measurements, temperature stability e.g. what motivates your manuscript.

*Calibration runs:*

Are calibration and data runs performed at the same relative humidity? (Think of cloud formation, latent heat release...)

Are there calibration runs that were performed at data run like flow rate? You could simply let the instruments suck as well – higher flow rate might increase the temperature instability. Thus, it would be necessary to show. How are clouds formed in a calibration run? If clouds are only formed in expansions, what is the meaning of calibration runs for cloud studies in the chamber?

You only show examples of the measurements in the figures for the calibration campaign. Is there a way of showing all data in one plot? Are there any expansions made during the calibration campaign? As the calibration strings were only installed during the calibration campaign, this would need to mimic conditions as they would be during a measurement campaign. If not, what is the aim of the calibration campaign?

You state e.g. in line 148 that around 300 expansions have been performed in the latter campaign. I would expect something like scatterplots showing all data and median/average values (if necessary grouped into classes by speed of expansions to show all data. "Various experimental conditions" are mentioned in the abstract, not mentioned any further later on! What are these various conditions? They could be used to group data for plots. Where is the statistical analysis? What about significances?

*Lab calibration:*

You mention that the WIKA reference thermometer is calibrated in the temperature range 0-100°C. How to you use it at temperatures below freezing? What confidence do you have in its performance there?

You mention liquid nitrogen as calibration point for cold temperatures. Is it valid to assume linear calibration between -196.21°C and 0°C, why? How did you get calibration points at temperatures between -70°C and 0°C, i.e. at temperatures that would potentially be used for experiments in the chamber? (E.g. you could add one point by using

salt/ice mix and one point by using dry ice in a Dewar flask). How did you calibrate at 0°C, this is not mentioned in the text? Figure A2 is showing a very different behaviour of the sensors at 0°C compared to the water bath calibration points (which, as you state, start at 2°C). So, how trustworthy is the point at 0°C?

Why are the OS, TC and Pt strings not calibrated directly in the lab as well? How exactly is linear interpolation performed for the Pt string (page 8, line 237/238)? Elaborate!

Apart from these main points, I found several typos and grammar errors that give the impression that the manuscript was not properly proof read before submission. Please be mindful of the reviewers, it greatly enhances readability when a manuscript is properly proof read – reviewers are not copy editors.

**Further comments:**
*Abstract:*
Please, be precise!
Line 11: lower flow rate of 20l/min stated here, 10l/min stated later in the manuscript (line 68) – what is correct? Also, 210l/min said here but 200l/min (p6, line 182) in the conclusions.
Line 16: "a few times 0.01°C" – exactly how many degrees?
Line 18: "larger non-uniformities" – exactly how big?

*Introduction:*
Line 31/32: move "is added" to the end of the sentence.
Line 35: Maybe a reference to the CERN Proton Synchroton?
Line 41: space missing before and point missing after "(Voigtlander et al., 2012)".
Line 42/43: "... (the small excess ensures..." better continue " that no contaminant vapours enter..."
Line 44: Wilson chamber – is there a reference?

*Section 2:*
Line 70-74: As mentioned earlier, a schematic of the chamber showing the positions of the serpentine pipe and the inlets where the warm trace gases are injected would be desirable.
Line 78: space missing before "(Kirkby...)"

*Section 3:*
Line 112: "short term ($>$15s)" should be "$<$15s"?
Line 126: "experimental hall temperature" – wall temperature?
Line 129: "high flow flow" – remove one "flow"

*Section 4:*
Line 147: "The characteristic reheating time constant, .." – a verb is missing.
Line 167/170: References to Fig. 6c and Fig. 6d should be Fig. 10c and Fig. 10d?

*Conclusions:*
Line 181 "high flow flow"... as earlier, remove one flow.

*Appendix:*
This describes the calibration in detail. Isn't the temperature calibration a main point in this study? In that case, it should be found in the main manuscript.
Line 203: What is the uncertainty at temperatures below freezing?
Line 223: Figure A2 shows a calibration point at 0°C, however, you do not mention how you calibrated at this point (water bath from 2°C on)?

*Figures:*
Figure 1: typo - dessicated = desiccated
Check flow rates (here 20l/min and 210l/min)

Figure 4: How did you choose which sensors you show here? Motivate your choice. You could also show all other sensors in a supplement.

Figure 6: Mark which data points result from the calibration runs and which from data

campaign. What does 21% and 100% fan mean, please explain.

Figure 7: Is the right hand axis really showing pressure, or rather $\delta$p (i.e. overpressure in the chamber)?

Figure 10: same as Fig. 7.

Figure A4: You mention a malfunction of the PTH2 sensor, was this true for the whole campaign?

Figure A5: Were the OS4 and OS5 sensors replaced after the calibration campaign (as obviously they showed an unusual behaviour)?

*Tables:* Table 1: From this table one could think that the TC and PTH / OS and PTV sensors have the same position. It would be better to indicate the offset in the table, e.g. by saying "0 (-20)" in the Height column (TC and PTH), and accordingly for the radius column for the OS and PTV sensors.

Table A4: Why are PT4 and PT6 missing?

I recommend a thorough overhaul of the manuscript before resubmission and potential publication.

---

## Author Comment (AC1) · 10 Jul 2017

We would like to thank the reviewer for taking the time to review the presented manuscript and provide their appraisal of the research carried out. We will try to provide a satisfactory reply to all the questions raised in this document. The posed questions are in green and the replies are presented below them.

Before replying to the individual questions we would like to offer our apologies for the amount of typos found in the manuscript. A complete evaluation of the manuscript has been carried out again and both the issues presented by the reviewer and others found during this new evaluation have been addressed and we hope the new manuscript will present a clearer and more understandable text.

**Major Questions**

Question 1: After reading the manuscript several times, now have the impression that during the calibration campaign only runs with steady flow conditions have been performed. Is this true? If so, why did you not perform evacuations for cloud formation as well? How much data (e.g. in hours, or if evacuations how many) did you take during the calibration campaign. How many hours did you measure in flow conditions during the data campaign? I think this information would be useful and should be added to the manuscript. Please clarify! In general, more information about the experiments performed during both campaigns might be helpful, maybe summarise the experiments in a table?

The calibration campaign stated in the manuscript did indeed only consist of measurements taken during steady state conditions. No evacuations were preformed. The reason for the calibration campaign was to create a calibration curve for the different permanent temperature strings that exist in CLOUD (the Pt100, TC and OS strings). In order to do this, several points at different operational temperatures in CLOUD are required. One temperature measurement is thus required to be acquired at stable conditions. Although expansions (evacuations) provide important information about the dynamics of the chamber, they do not provide information relevant to obtaining a calibration curve of the sensors. Each temperature point taken during the calibration campaign took 24 hours. During data campaigns the permanent sensor strings (Pt100, TC, OS) were constantly turned on and acquiring data. The relevant nucleation experiments used were taken during separate periods between 29th of September and 29th of October 2014. Each temperature point during data campaigns consists of a period no smaller than 3 hours. The manuscript has been altered to better relay this information and the focus of these experiments and a table containing the major relevant parameters of these nucleation experiments has been included.

Question 2: Why Pt100 sensors were chosen for the calibration strings? This is not motivated in the manuscript. However, as you state, they have a rather long response time (180 s), which seems insufficient for cloud experiments, where temperature drops much faster than this? How do you compare temperatures

from the fast response sensors to these slow response sensors? Please explain!

The manuscript was indeed confusing regarding this point. A distinction must be made between the permanent Pt100 string and the calibration PT100 strings (PTH and PTV). The description of the Pt100 string and more specifically the 180s response time was regarding the permanent Pt100 string. The calibration Pt100 strings use a completely different hardware and software construction, allowing for 1s time resolution measurements and smaller time response. A revised paragraph on the construction of the PT100 calibration strings was added, containing the following text distinguishing the Pt100 strings:

"These sensors, unlike the Pt100 string already present in the CLOUD chamber and despite also being four-wire sensors with National Instruments NI~9217 readout electronics, do not have the high mass of the first sensors, allowing for fast responses equal to the TC and OS sensors."

Question 3: Why do you average the data over 15sec? Does it make sense in case of the Pt100 with time constants around 180sec? On the other hand, 15secs smooth out fluctuations in the fast sensors responses. Do you also smooth data during evacuations?

The data in the plots shown in figures 3 and 4 was altered using a median filter. This filter applies a median to a specified window of data (in this case 15 seconds) as opposed to the averaging mechanism suggested by the reviewer that applies an average instead. The reason for this data alteration is two-fold. First, to remove any outliers from the measurement of the distribution of temperature. Secondly to provide the reader with an easier to read figure, since without the filter, the reader would not be able to see the fluctuations of the data and the different signals shown. Indeed, temperature drops during cloud formation experiments (expansions) were at times much faster than 180 seconds. The temperature measurements shown in this manuscript for expansions, however, do not show measurements from the permanent Pt100 string, only the fast response OS and TC strings. The permanent Pt100 string only exists to measure nucleation experiments, the comparison of data from this string to the fast response sensors is thus valid, taking into account the total run time of nucleation experiments (several hours). The data for expansions was not altered since we were not measuring a constant temperature distribution but a time-changing one. In every instance where data was altered, it is explained to the reader in the manuscript. The following text was added to the caption of figure 3:
"The measurements are smoothed with a 15 s median window firstly to improve the measurement result by removing any outlier values of the measurement distribution and secondly to improve readability of the figure by the reader."

Question 4: I would like to see a schematic of CLOUD chamber that shows more detail, e.g. the "serpentine" pipe (page 3, line 71), regulation and gate valves (line 51), sampling ports for cloud measurements. Particularly the valve positions in respect to the temperature sensors would be good to know!

The gas input pipes are located below the lower mixing fan, which is then responsible for pulling the inputted gas and mixing it with the air inside the chamber. They insert the gases at a height of a couple of centimeters above the bottom of the chamber (-150 cm < z <-140 cm using figure1) Figure 1 has been altered to include a representation of the gas input valves along with a representation of the axis used to calculate the sensor positions in the chamber. The reader can now estimate the position of the pipes in relation to the temperature sensors (using figure 1 and table 1).

Question 5: A large part of the introduction (i.e. from line 28) reads more like a potential chapter 2 "chamber operation" (or similar). I would expect more introduction about cloud chambers and the importance of temperature measurements, temperature stability e.g. what motivates your manuscript.

The manuscript has been changed to reflect the expressed views. Section 2 is now named chamber operation and part of the offending text in the introduction was moved there. The following text was included to stress the relevance of temperature characterization of chamber measurements:
"Maintaining temperature stability and uniformity in these chamber measurements ensures that the chemical reaction rates in the chamber do no fluctuate either in time or in space [2]. Accurate measurement of temperature is also necessary to measure the onset of ice formation in chamber experiments [3, 1]."
(citations properly introduced in manuscript)

**Calibration Runs**

Question 1: Are calibration and data runs performed at the same relative humidity? (Think of cloud formation, latent heat release...)

No dedicated measurement of relative humidity was taken during the calibration runs. However, the proximity of the calibration sensors to the TC and OS sensors as well as the sheer length of measurements during the calibration campaign drastically reduces the uncertainty caused by varied values of relative humidity. The campaign data was also taken at various relative humidity values. The nucleation experiments were a part of scheduled experiments of one of the many institutes that are a part of the CLOUD consortium and were not subject to any change. There were no dedicated experiments during the data campaign for temperature measurements (either nucleation or expansion experiments). The manuscript has been altered to reflect this concern.

Question 2: Are there calibration runs that were performed at data run like flow rate? You could simply let the instruments suck as well – higher flow rate might increase the temperature instability. Thus, it would be necessary to show. How are clouds formed in a calibration run? If clouds are only formed in expansions, what is the meaning of calibration runs for cloud studies in the chamber?

The instruments referred to in the manuscript and this question were not yet present at the time in CLOUD. The instruments present in CLOUD data

campaigns are the property of several institutes in the CLOUD consortium (and outside it as well). These instruments are taking measurements at several sites around the world during the year. Only at specific times are they present at CLOUD. At the time of the calibration campaign, no instrument had yet arrived. The very high cleanliness standards of CLOUD also prevent us from keeping instrument ports open for large periods of time. Thus it was simply impossible to mimic the flow conditions of CLOUD data campaigns without inadvertently increasing the pressure in the chamber. No clouds were formed in the calibration run. As stated in the first question, the goal of the calibration campaign was to provide a calibration curve for the temperature sensors. No other instruments were calibrated during this campaign. The manuscript was altered to clarify the definition of the calibration campaign throughout the text.

Question 3: You only show examples of the measurements in the figures for the calibration campaign. Is there a way of showing all data in one plot? Are there any expansions made during the calibration campaign? As the calibration strings were only installed during the calibration campaign, this would need to mimic conditions as they would be during a measurement campaign. If not, what is the aim of the calibration campaign?

Figure 3a, 6a, 6b, 7 and 10 show data taken during CLOUD data campaigns. If the reviewer is referring to the measurement distributions provided in figure 4, this can be remedied. It would however, require the reviewers clarification. No expansions were made during the calibration campaign as explained in the first and previous question. See previous question for reason of calibration campaign

Question 4: You state e.g. in line 148 that around 300 expansions have been performed in the latter campaign. I would expect something like scatterplots showing all data and median/average values (if necessary grouped into classes by speed of expansions to show all data. "Various experimental conditions" are mentioned in the abstract, not mentioned any further later on! What are these various conditions? They could be used to group data for plots. Where is the statistical analysis? What about significances?

Over 300 expansions were indeed preformed in the CLOUD campaign. An indirect representation of the distribution of expansions is presented in figure 9 accompanied with statistical analysis. The temperature change in an expansion is indirectly related to the time of the expansion. The variation of experimental parameters is too large to provide an easy visualization of the parameters. Presenting the different expansions also goes outside the scope of the manuscript, as only the relevant dynamic behavior of expansions in the CLOUD chamber is studied in this manuscript. A statistical analysis is provided with the use of the expansion reheating parameters in figures 9 and 8. The reheating parameter in our opinion provides a quantitative parameter to analyze individual expansions.

**Lab Calibration**

Question 1: You mention that the WIKA reference thermometer is calibrated in the temperature range 0-100∘C. How to you use it at temperatures below freezing? What confidence do you have in its performance there?

The WIKA thermometer was only used for the positive temperature values, including the 0ºC point. The appendix has been edited to clarify this point.

Question 2: You mention liquid nitrogen as calibration point for cold temperatures. Is it valid to assume linear calibration between -196.21∘C and 0∘C, why? How did you get calibration points at temperatures between -70∘C and 0∘C, i.e. at temperatures that would potentially be used for experiments in the chamber? (E.g. you could add one point by using salt/ice mix and one point by using dry ice in a Dewar flask).

As expressed in the manuscript, the Pt100 sensors were calibrated using the Callendar-van Dusen equation. This equation provides a curve for resistance of Pt100 sensors in the range of temperature of [-200 ºC,600 ºC]. This equation is a quadratic equation for temperatures above 0º C and a third degree equation for temperatures lower than 0 ºC. This relationship is based on the properties of platinum that are part of the sensors. There were no laboratory temperature points in the range of CLOUD's negative temperatures especially because the reference thermometer was not absolutely calibrated below 0º C. The Callendar-van Dusen equation along with the estimated parameters will provide a relationship between each sensor's resistance and temperature in the whole range of CLOUD operating temperatures. The uncertainty of the measurements of the Pt100 sensors was estimated on a worst case scenario using the Monte Carlo study for the ranges of temperatures in CLOUD (including the negative temperatures). The manuscript was altered to clarify first the use of the Callendar-van Dusen equation and secondly the importance of the Monte Carlo study in the calculation of the temperature uncertainty for points without laboratory measurements.

The suggested salt-water mixture suggested by the author was considered but the use of said mixture in the laboratory would go against the strict cleanliness standards of the CLOUD experiment, running the risk of contaminating further campaigns after placing the strings in the chamber. The following text was added to the appendix:

"In order to comply with CLOUD's cleanliness standards, only pure water and liquid nitrogen were allowed for the laboratory calibration of the Pt100 sensors."

Question 3: How did you calibrate at 0 ∘C, this is not mentioned in the text? Figure A2 is showing a very different behaviour of the sensors at 0∘C compared to the water bath calibration points (which, as you state, start at 2∘C). So, how trustworthy is the point at 0∘C?

The calibration at 0ºC was accomplished by using a mixture of milipore water and ice. During this phase transition it is guaranteed that the temperature is at 0ºC. There are, however some concerns to take into account, mostly

related to the volume of the Huber unit used. The most important of which is the possible existence of convection currents that increase the uncertainty of the measurement. The following text has been added describing the $0^{\circ}$ C point:

"A 0°C point was obtained using a millipore water bath containing ice, ensuring that the temperature was at 0°C while ice was present in the bath."

Question 4: Why are the OS, TC and Pt strings not calibrated directly in the lab as well? How exactly is linear interpolation performed for the Pt string (page 8, line 237/238)? Elaborate!

The reason for the creation of these specialized strings was to make an in-situ calibration of the permanent temperature sensors. It was noticed that when the sensors of the permanent strings were disconnected from their readouts, their calibrations changed. This makes it impossible to disconnect the strings from the chamber and make a proper calibration only to reconnect the readouts and find out that the calibration had changed. These specialized strings were made in such a way that the string could be unmounted and mounted in the chamber while keeping the individual sensors connected to their respective readouts, allowing for a laboratory calibration and subsequent in-situ chamber calibration. The manuscript now stresses this point throughout with passages such as the following which has been added to the end of section 2 to calrify this point:

"Removing the sensor strings to calibrate requires that the sensors be disconnected from their respective electronics, this results in a shift of the previous calibration. It is thus impossible to remove the strings for calibration and place them back in the chamber using the same calibration."

In response to the concern as to how the interpolation of the sensor was made. This was a spatial linear interpolation of the temperature between the position of PTH sensors and the permanent Pt100 sensors. The following text was added to the manuscript as an effort to explain how the interpolation was made for the also non working PTH2 (see question 4 in next section), which also applies for the permanent Pt100 string and this has been stressed in the manuscript:

"PTH2 was non responsive after being placed in the chamber. Thus it's measurement was replaced by a spatial linear fit between the measurements of PT1H and PT3H defined by:

$$T^*_{PTH2} = T_{PTH1} + (r_{PTH2} - r_{PTH1})\frac{T_{PTH3} - T_{PTH1}}{r_{PTH3} - r_{PTH1}} \qquad (1)$$

where $r_{PTY}$ is the radial position of sensor PTHY and $T_{PTHY}$ is the temperature measured by sensor PTHY (see table 1 for details)."

**Other questions - which were not answered in other sections**

Question 1: "experimental hall temperature" – wall temperature?

This indeed refers to the hall temperature at CERN where the CLOUD chamber is placed. Since it is a large hall shared by many other experiments, not to mention provides access to all scientists in and out of CLOUD to (verify instrumentation, etc..), it is simply impossible to provide temperature control for this hall at CLOUD operational temperatures). The feed-through gas pipes travel a short path (a few meters) through this hall before going either through the temperature controlled serpentine cable or being directly sent into CLOUD and there is an inevitable alteration of temperature during this path.

**Question 2: Appendix: This describes the calibration in detail. Isn't the temperature calibration a main point in this study?**

This question was also considered by the authors. The decision was to provide an appendix to the paper due to the fact that the manuscript is related to the analysis of the temperature stability, uniformity of the chamber. While the calibration of the sensors and how it was done is indeed important to show, it should not overshadow the end goal of the manuscript.

**Question 3: Figure 4: How did you choose which sensors you show here? Motivate your choice. You could also show all other sensors in a supplement.**

The sensors were chosen as they are the sensors at the middle of each respective string. A supplement has been prepared with the measurements of all other sensors in the string at the respective temperature, containing the requested plots, which are shown at the end of this document. The following text was added to figure 4:

"OS3, TC3 and PT3 were chosen due to being the central sensors of each respective string. The temperatures chosen represent common temperatures used during CLOUD campaigns."

**Question 4: Figure A4: You mention a malfunction of the PTH2 sensor, was this true for the whole campaign?**

This was indeed true during the whole calibration campaign. This was resolved in data analysis by creating a virtual sensor via a spatial linear interpolation of the temperature between the radial position of PT1H (145 cm) and PT3H (90 cm) much as the same used for calibrating the permanent Pt100 string. An explanation of the interpolation mechanism can be found in question 4 of the previous section and as that question states, the manuscript has been updated.

**Question 5: Were the OS4 and OS5 sensors replaced ater the calibration campaign (as obviously they showed an unusual behaviour)?**

The sensors were not replaced, as doing so would involve removing all other sensors from their respective readouts. This has been clarified in the manuscript.

**Question 6: Table 1: From this table one could think that the TC and PTH / OS and PTV sensors have the same position. It would be better to indicate**

the offset in the table, eg. by saying "0 (-20)" in the Height column (TC and PTH), and accordingly for the radius column for the OS and PTV sensors.

The sensors were placed in their strings and installed in the CLOUD chamber in an effort to place them in the closest possible position to each sensor to be calibrated. No measurement was made to determine the offset, visual inspection concluded that the sensors were offset by no more than 1 cm. As suggested the values PT $\pm$ 1 cm were added to the PTH and PTV sensor positions in table 1 to indicate that their positions did not exactly match.

Question 7: Table A4: Why are PT4 and PT6 missing?

The permanent Pt100 string is only a 5 sensor string. PT4 was not functioning at the time of the calibration campaign and data campaign. This malfunction occurred during one of many efforts to calibrate said string in lab. One of such efforts must have damaged the sensing tip. During these efforts we noticed the shift in calibration when reconnecting the electronics which led us to make the calibration process described in the manuscript. The manuscript has been altered to address this concern.

We again would like to thank the reviewer for the time taken to appraise this manuscript. We hope that these replies along with the provided supplemented material and changes to the manuscript can provide a more favorable recommendation.

**References**

[1] PJ Connolly, C Emersic, and PR Field. A laboratory investigation into the aggregation efficiency of small ice crystals. *Atmospheric Chemistry and Physics*, 12(4):2055–2076, 2012.

[2] Howard G. Maahs. Kinetics and mechanism of the oxidation of s(IV) by ozone in aqueous solution with particular reference to SO2conversion in nonurban tropospheric clouds. *Journal of Geophysical Research*, 88(C15):10721, 1983.

[3] Birte Riechers, Frank Wittbracht, Andreas Hütten, and Thomas Koop. The homogeneous ice nucleation rate of water droplets produced in a microfluidic device and the role of temperature uncertainty. *Physical Chemistry Chemical Physics*, 15(16):5873, 2013.

---

## Referee Comment (RC2) · Anonymous Referee #2 · 8 Aug 2017

The manuscript describes a set of laboratory studies of the temperature behaviour in a cloud chamber.

Temporal, spatial and chamber temperature dependence of the thermometric measurements are investigated.

Temporal and spatial variations in the temperature measurements are as little as 0.01 C but can be as much as 0.1 C when the chamber temperature is varied between -70 C and +40 C.

These are impressive temperature measurements.

The technique of calibration for the thermal sensors is a benchmark for any cloud chamber.

[Figure]

As an instrumentation paper, this work is of great value.

It may be necessary to illustrate one or two examples of scientific studies one could do with such high precision measurement in the CERN CLOUD chamber.

The authors have mentioned in the text the importance of precise temperature measurements but a concrete example of an experiment that has been carried out in the CLOUD chamber will be more convincing to the readers.

In general the paper is well written although some amount of polishing on the editorial side will improve the manuscript.

I recommend publication of this manuscript with minor modifications.

---

## Author Response (AR2)

**Description of manuscript changes**

**General changes**

Throughout the manuscript, several sentences were altered to increase readability and address some concerns pointed out by reviewer #1. Several changes were made to the manuscript after the replies to reviewer #1 were made, thus, the provided replies are not necessarily up to date. A better description and differentiation of the different Pt100 strings was made. References to the different periods of data acquisition were strengthened. There are now clear indications in the image captions as to whether there was any smoothing of data. The introduction was reorganized to provide a proper introduction to cloud chambers, their importance and the specific importance of temperature measurements for new particle formation (NPF) experiments and cloud formation experiments. The laboratory calibration of sensors was also altered to clarify several points which were less understandable

**Specific changes**

**Introduction**

Three paragraphs were introduced. Firstly, highlighting the scientific relevance of tank reactors in the study of atmospheric processes and NPF, along with the contributions of the CLOUD chamber to these studies. Secondly, highlighting the importance of expansion chambers in the study of in-cloud processes as well as the contributions of the CLOUD chamber to this particular field of study. Thirdly a motivation on the importance of temperature measurement and stability in the study of NPF and in-cloud processes.

**NEW - Chamber operation**

A second section was added to describe the general operation of the CLOUD chamber, both during NPF studies and in-cloud process studies.

**Thermal system**

A third section now goes into specific detail on the thermal control system and temperature sensors available in CLOUD. Figure 1 was updated to include the

position of the trace gas input lines. Table 1 was updated to include an estimate of the position of the calibration Pt100 sensor strings.

**Results - NPF and in-cloud process experiments**

The sections on the results obtained during NPF and in-cloud process experiments were largely altered to increase readability, a table was added describe the different NPF experiments used to acquire data.

**Appendix A**

Annex A was altered, mostly to increase readability.

**Editor replies - 02 Nov 2017**

References to the supplementary material were added to the manuscript.

**Replies to reviewer #1**

The replies to reviewer #1 have been largely kept the same as the ones provided in the interactive discussion on the AMT website. Some replies were edited to reflect the current status of the manuscript. As some of the paragraphs that were said to have been placed either suffered changes or were removed in favor of readability

**Major Questions**

Question 1: After reading the manuscript several times, now have the impression that during the calibration campaign only runs with steady flow conditions have been performed. Is this true? If so, why did you not perform evacuations for cloud formation as well? How much data (e.g. in hours, or if evacuations how many) did you take during the calibration campaign. How many hours did you measure in flow conditions during the data campaign? I think this information would be useful and should be added to the manuscript. Please clarify! In general, more information about the experiments performed during both campaigns might be helpful, maybe summarize the experiments in a table?

The calibration campaign stated in the manuscript did indeed only consist of measurements taken during steady state conditions. No evacuations were preformed. The reason for the calibration campaign was to create a calibration curve for the different permanent temperature strings that exist in CLOUD (the Pt100, TC and OS strings). In order to do this, several points at different operational temperatures in CLOUD are required. One temperature measurement is thus required to be acquired at stable conditions. Although expansions (evacuations) provide important information about the dynamics of the chamber, they do not provide information relevant to obtaining a calibration curve of the sensors. Each temperature point taken during the calibration campaign took 24 hours. During data campaigns the permanent sensor strings (Pt100, TC, OS) were constantly turned on and acquiring data. The relevant nucleation experiments used were taken during separate periods between 29th of September and 29th of October 2014. Each temperature point during data campaigns consists of a period no smaller than 3 hours. The manuscript has been altered to better relay this information and the focus of these experiments and a table containing the major relevant parameters of these nucleation experiments has been included.

Question 2: Why Pt100 sensors were chosen for the calibration strings? This is not motivated in the manuscript. However, as you state, they have a rather long response time (180 s), which seems insufficient for cloud experiments, where temperature drops much faster than this? How do you compare temperatures from the fast response sensors to these slow response sensors? Please explain!

The manuscript was indeed confusing regarding this point. A distinction must be made between the permanent Pt100 string and the calibration PT100 strings (PTH and PTV). The description of the Pt100 string and more specifically the 180s response time was regarding the permanent Pt100 string. The calibration Pt100 strings use a completely different hardware and software construction, allowing for 1s time resolution measurements and smaller time response.

Question 3: Why do you average the data over 15sec? Does it make sense in case of the Pt100 with time constants around 180sec? On the other hand, 15secs smooth out fluctuations in the fast sensors responses. Do you also smooth data during evacuations?

The data in the plots shown in figures 3 and 4 was altered using a median filter. This filter applies a median to a specified window of data (in this case 15 seconds) as opposed to the averaging mechanism suggested by the reviewer that applies an average instead. The reason for this data alteration is two-fold. First, to remove any outliers from the measurement of the distribution of temperature. Secondly to provide the reader with an easier to read figure, since without the filter, the reader would not be able to see the fluctuations of the data and the different signals shown. Indeed, temperature drops during cloud formation experiments (expansions) were at times much faster than 180 seconds. The temperature measurements shown in this manuscript for expansions, however, do not show measurements from the permanent Pt100 string, only the fast response OS and TC strings. The permanent Pt100 string only exists to measure nucleation experiments, the comparison of data from this string to the fast response sensors is thus valid, taking into account the total run time of nucleation experiments (several hours). The data for expansions was not altered since we were not measuring a constant temperature distribution but a time-changing one. In every instance where data was altered, it is explained to the reader in the manuscript.

Question 4: I would like to see a schematic of CLOUD chamber that shows more detail, e.g. the "serpentine" pipe (page 3, line 71), regulation and gate valves (line 51), sampling ports for cloud measurements. Particularly the valve positions in respect to the temperature sensors would be good to know!

The gas input pipes are located below the lower mixing fan, which is then responsible for pulling the inputted gas and mixing it with the air inside the chamber. They insert the gases at a height of a couple of centimeters above the

bottom of the chamber (-150 cm < z <-140 cm using figure1) Figure 1 has been altered to include a representation of the gas input valves along with a representation of the axis used to calculate the sensor positions in the chamber. The reader can now estimate the position of the pipes in relation to the temperature sensors (using figure 1 and table 1).

Question 5: A large part of the introduction (i.e. from line 28) reads more like a potential chapter 2 "chamber operation" (or similar). I would expect more introduction about cloud chambers and the importance of temperature measurements, temperature stability e.g. what motivates your manuscript.

The manuscript has been changed to reflect the expressed views. Section 2 is now named chamber operation and part of the offending text in the introduction was moved there. The introduction was increased to include the relevance of cloud chambers in the study of NPF and in-cloud processes, CLOUD's contribution to these studies and the specific importance of temperature measurements in the CLOUD chamber.

**Calibration Runs**

Question 1: Are calibration and data runs performed at the same relative humidity? (Think of cloud formation, latent heat release...)

No dedicated measurement of relative humidity was taken during the calibration runs. However, the proximity of the calibration sensors to the TC and OS sensors as well as the sheer length of measurements during the calibration campaign drastically reduces the uncertainty caused by varied values of relative humidity. The campaign data was also taken at various relative humidity values. The nucleation experiments were a part of scheduled experiments of one of the many institutes that are a part of the CLOUD consortium and were not subject to any change. There were no dedicated experiments during the data campaign for temperature measurements (either nucleation or expansion experiments). The manuscript has been altered to reflect this concern.

Question 2: Are there calibration runs that were performed at data run like flow rate? You could simply let the instruments suck as well − higher flow rate might increase the temperature instability. Thus, it would be necessary to show. How are clouds formed in a calibration run? If clouds are only formed in expansions, what is the meaning of calibration runs for cloud studies in the chamber?

The instruments referred to in the manuscript and this question were not yet present at the time in CLOUD. The instruments present in CLOUD data campaigns are the property of several institutes in the CLOUD consortium (and outside it as well). These instruments are taking measurements at several sites around the world during the year. Only at specific times are they present at

CLOUD. At the time of the calibration campaign, no instrument had yet arrived. The very high cleanliness standards of CLOUD also prevent us from keeping instrument ports open for large periods of time. Thus it was simply impossible to mimic the flow conditions of CLOUD data campaigns without inadvertently increasing the pressure in the chamber. No clouds were formed in the calibration run. As stated in the first question, the goal of the calibration campaign was to provide a calibration curve for the temperature sensors. No other instruments were calibrated during this campaign. The manuscript was altered to clarify the definition of the calibration campaign throughout the text.

Question 3: You only show examples of the measurements in the figures for the calibration campaign. Is there a way of showing all data in one plot? Are there any expansions made during the calibration campaign? As the calibration strings were only installed during the calibration campaign, this would need to mimic conditions as they would be during a measurement campaign. If not, what is the aim of the calibration campaign?

Figure 3a, 6a, 6b, 7 and 10 show data taken during CLOUD data campaigns. If the reviewer is referring to the measurement distributions provided in figure 4, this can be remedied. It would however, require the reviewers clarification. No expansions were made during the calibration campaign as explained in the first and previous question. See previous question for reason of calibration campaign

Question 4: You state e.g. in line 148 that around 300 expansions have been performed in the latter campaign. I would expect something like scatterplots showing all data and median/average values (if necessary grouped into classes by speed of expansions to show all data. "Various experimental conditions" are mentioned in the abstract, not mentioned any further later on! What are these various conditions? They could be used to group data for plots. Where is the statistical analysis? What about significances?

Over 300 expansions were indeed preformed in the CLOUD campaign. An indirect representation of the distribution of expansions is presented in figure 9 accompanied with statistical analysis. The temperature change in an expansion is indirectly related to the time of the expansion. The variation of experimental parameters is too large to provide an easy visualization of the parameters. Presenting the different expansions also goes outside the scope of the manuscript, as only the relevant dynamic behavior of expansions in the CLOUD chamber is studied in this manuscript. A statistical analysis is provided with the use of the expansion reheating parameters in figures 9 and 8. The reheating parameter in our opinion provides a quantitative parameter to analyze individual expansions.

**Lab Calibration**

Question 1: You mention that the WIKA reference thermometer is calibrated in the temperature range 0-100∘C. How to you use it at temperatures below

freezing? What confidence do you have in its performance there?

The WIKA thermometer was only used for the positive temperature values, including the 0ºC point. The appendix has been edited to clarify this point.

Question 2: You mention liquid nitrogen as calibration point for cold temperatures. Is it valid to assume linear calibration between -196.21∘C and 0∘C, why? How did you get calibration points at temperatures between -70∘C and 0∘C, i.e. at temperatures that would potentially be used for experiments in the chamber? (E.g. you could add one point by using salt/ice mix and one point by using dry ice in a Dewar flask).

As expressed in the manuscript, the Pt100 sensors were calibrated using the Callendar-van Dusen equation. This equation provides a curve for resistance of Pt100 sensors in the range of temperature of [-200 ºC,600 ºC]. This equation is a quadratic equation for temperatures above 0º C and a third degree equation for temperatures lower than 0 ºC. This relationship is based on the properties of platinum that are part of the sensors. There were no laboratory temperature points in the range of CLOUD's negative temperatures especially because the reference thermometer was not absolutely calibrated below 0º C. The Callendar-van Dusen equation along with the estimated parameters will provide a relationship between each sensor's resistance and temperature in the whole range of CLOUD operating temperatures. The uncertainty of the measurements of the Pt100 sensors was estimated on a worst case scenario using the Monte Carlo study for the ranges of temperatures in CLOUD (including the negative temperatures). The manuscript was altered to clarify first the use of the Callendar-van Dusen equation and secondly the importance of the Monte Carlo study in the calculation of the temperature uncertainty for points without laboratory measurements.
The suggested salt-water mixture suggested by the author was considered but the use of said mixture in the laboratory would go against the strict cleanliness standards of the CLOUD experiment, running the risk of contaminating further campaigns after placing the strings in the chamber. The manuscript was altered to clarify this point:

Question 3: How did you calibrate at 0 ∘C, this is not mentioned in the text? Figure A2 is showing a very different behaviour of the sensors at 0∘C compared to the water bath calibration points (which, as you state, start at 2∘C). So, how trustworthy is the point at 0∘C?

The calibration at 0ºC was accomplished by using a mixture of milipore water and ice. During this phase transition it is guaranteed that the temperature is at 0ºC. There are, however some concerns to take into account, mostly related to the volume of the Huber unit used. The most important of which is the possible existence of convection currents that increase the uncertainty of the measurement. The manuscript was altered to clarify this point.

Question 4: Why are the OS, TC and Pt strings not calibrated directly in the lab as well? How exactly is linear interpolation performed for the Pt string (page 8, line 237/238)? Elaborate!

The reason for the creation of these specialized strings was to make an in-situ calibration of the permanent temperature sensors. It was noticed that when the sensors of the permanent strings were disconnected from their readouts, their calibrations changed. This makes it impossible to disconnect the strings from the chamber and make a proper calibration only to reconnect the readouts and find out that the calibration had changed. These specialized strings were made in such a way that the string could be unmounted and mounted in the chamber while keeping the individual sensors connected to their respective readouts, allowing for a laboratory calibration and subsequent in-situ chamber calibration. The manuscript now stresses this point throughout with passages such as the following which has been added to the end of section 2 to clarify this point:

"Removing the OS, TC and PT sensor strings for calibration would require disconnection from their readout electronics, which can result in a shift of their calibration."

In response to the concern as to how the interpolation of the sensor was made. This was a spatial linear interpolation of the temperature between the position of PTH sensors and the permanent Pt100 sensors. The following text was added to the manuscript as an effort to explain how the interpolation was made for the also non working PTH2 (see question 4 in next section), which also applies for the permanent Pt100 string and this has been stressed in the manuscript:

"PTH2 was non responsive after being placed in the chamber. Thus its measurement was replaced by a spatial linear fit between the measurements of PTH1 and PTH3 defined by::

$$T^*_{PTH2} = T_{PTH1} + (r_{PTH2} - r_{PTH1}) \frac{T_{PTH3} - T_{PTH1}}{r_{PTH3} - r_{PTH1}} \qquad (1)$$

where $r_{PTHi}$ and $T_{PTHi}$ are, respectively, the radial position and temperature measured by the PTH sensor of index $i$, as defined in Table 1.

**Other questions - which were not answered in other sections**

Question 1: "experimental hall temperature" − wall temperature?

This indeed refers to the hall temperature at CERN where the CLOUD chamber is placed. Since it is a large hall shared by many other experiments, not to mention provides access to all scientists in and out of CLOUD to (verify instrumentation, etc..), it is simply impossible to provide temperature control for this hall at CLOUD operational temperatures). The feed-through gas pipes

travel a short path (a few meters) through this hall before going either through the temperature controlled serpentine cable or being directly sent into CLOUD and there is an inevitable alteration of temperature during this path.

**Question 2: Appendix: This describes the calibration in detail. Isn't the temperature calibration a main point in this study?**

This question was also considered by the authors. The decision was to provide an appendix to the paper due to the fact that the manuscript is related to the analysis of the temperature stability, uniformity of the chamber. While the calibration of the sensors and how it was done is indeed important to show, it should not overshadow the end goal of the manuscript.

**Question 3: Figure 4: How did you choose which sensors you show here? Motivate your choice. You could also show all other sensors in a supplement.**

The sensors were chosen as they are the sensors at the middle of each respective string. A supplement has been prepared with the measurements of all other sensors in the string at the respective temperature, containing the requested plots, which are shown at the end of this document.

**Question 4: Figure A4: You mention a malfunction of the PTH2 sensor, was this true for the whole campaign?**

This was indeed true during the whole calibration campaign. This was resolved in data analysis by creating a virtual sensor via a spatial linear interpolation of the temperature between the radial position of PT1H (145 cm) and PT3H (90 cm) much as the same used for calibrating the permanent Pt100 string. An explanation of the interpolation mechanism can be found in question 4 of the previous section and as that question states, the manuscript has been updated.

**Question 5: Were the OS4 and OS5 sensors replaced ater the calibration campaign (as obviously they showed an unusual behaviour)?**

The sensors were not replaced, as doing so would involve removing all other sensors from their respective readouts. This has been clarified in the manuscript.

**Question 6: Table 1: From this table one could think that the TC and PTH / OS and PTV sensors have the same position. It would be better to indicate the offset in the table, eg. by saying "0 (-20)" in the Height column (TC and PTH), and accordingly for the radius column for the OS and PTV sensors.**

The sensors were placed in their strings and installed in the CLOUD chamber in an effort to place them in the closest possible position to each sensor to be calibrated. No measurement was made to determine the offset, visual inspection

concluded that the sensors were offset by no more than 1 cm. As suggested the values PT $\pm$ 1 cm were added to the PTH and PTV sensor positions in table 1 to indicate that their positions did not exactly match.

Question 7: Table A4: Why are PT4 and PT6 missing?

The permanent Pt100 string is only a 5 sensor string. PT4 was not functioning at the time of the calibration campaign and data campaign. This malfunction occurred during one of many efforts to calibrate said string in lab. One of such efforts must have damaged the sensing tip. During these efforts we noticed the shift in calibration when reconnecting the electronics which led us to make the calibration process described in the manuscript. The manuscript has been altered to address this concern.

We again would like to thank the reviewer for the time taken to appraise this manuscript. We hope that these replies along with the provided supplemented material and changes to the manuscript can provide a more favorable recommendation.

**Replies to reviewer #2**

We would firstly like to thank the reviewer for the favorable appreciation of the manuscript. Below we address the requests brought forth by the reviewer. They follow the requested format by AMT. Reviewer requests are in green followed by the author's response.

Request 1: It may be necessary to illustrate one or two examples of scientific studies one could do with such high precision measurement in the CERN CLOUD chamber.

The manuscript was kept somewhat vague in regards to preformed experiments in order to focus on the quality and implications of the preformed measures. Several examples of studies preformed at CLOUD which require very accurate temperature control have been added to the introduction section.

Request 2: The authors have mentioned in the text the importance of precise temperature measurements but a concrete example of an experiment that has been carried out in the CLOUD chamber will be more convincing to the readers.

Two factors are weighted in the decision to not include specific experiment details. Firstly, we would like to stress the need to not bog the user down in experimental details that do not concern temperature measurements. Secondly, in every CLOUD campaign, several experiments are performed every year, such that, while some similarities exist, experiments are performed by different institutes and operators. As stated in the previous answer, several examples of different experiments preformed in CLOUD are now provided in the introduction section for the more curious readers.

**Replies - 02 Nov 2017**

In order to comply with the comments on the Editor's report, which suggested a link be added in the manuscript, connecting the supplementary material, we have altered the manuscript text in the following ways:

1. On line 152-153, the following sentence was added:
   "Supplemetary material provides distribution of residuals for all other sensors."

2. The caption of figure 4 was also altered to contain the following sentence:
   "See supplement material for distributions of other sensors."

Manuscript prepared for Atmos. Meas. Tech.
with version 2015/04/24 7.83 Copernicus papers of the LATEX class copernicus.cls.
Date: 2 November 2017

**Temperature uniformity in the CERN CLOUD chamber**

António Dias[1], Sebastian Ehrhart[1,a], Alexander Vogel[1,b], Christina Williamson[2,c], João Almeida[1,2], Jasper Kirkby[1,2], Serge Mathot[1], Samuel Mumford[1,d], and Antti Onnela[1]

[1]CERN, CH-1211 Geneva, Switzerland
[2]Goethe University Frankfurt, Institute for Atmospheric and Environmental Sciences, 60438 Frankfurt am Main, Germany
[a]Now at Max Planck Institute for Chemistry Atmospheric Chemistry Department
Hahn-Meitner-Weg 1, 55128 Mainz, Germany
[b]Now at Paul Scherrer Institute, Aarebrücke, 5232 Villigen, Switzerland
[c]Now at Chemical Sciences Division, NOAA Earth System Research Laboratory, Boulder, CO and CIRES, University of Colorado, Boulder, CO
[d]Now at Kapitulink Lab, 476 Lomita Mall, Stanford University, Stanford, CA 94305-4045

*Correspondence to:* António Dias (amcbd89@gmail.com)

**Abstract.** The CLOUD (Cosmics Leaving OUtdoor Droplets) experiment at CERN is studying the nucleation and growth of aerosol particles under atmospheric conditions, and their activation into cloud droplets. A key feature of the CLOUD experiment is precise control of the experimental parameters. Temperature uniformity and stability in the chamber are important since many of the pro-

5 cesses under study are sensitive to temperature and also to contaminants that can be released from the stainless steel walls by upward temperature fluctuations. The air enclosed within the  26 m$^3$ CLOUD chamber is equipped with several arrays ("strings") of high precision, fast-response thermometers to measure its temperature. Here we present a study of the air temperature uniformity inside the CLOUD chamber under various experimental conditions. Measurements were performed

10 under calibration conditions and run conditions, which are distinguished by the flow rate of fresh air and trace gases entering the chamber: 20 l/min and up to 210 l/min, respectively. During steady-state calibration runs between -70 °C and +20 °C, the air temperature uniformity is better than ±0.06 °C in the radial direction and ±0.1 °C in the vertical direction. Larger non-uniformities are present during experimental runs, depending on the temperature control of the make-up air and trace gases (since

15 some trace gases require elevated temperatures until injection into the chamber). The temperature stability  is ±0.04 °C over periods of several hours during either calibration or steady-state run conditions. During rapid adiabatic expansions to activate cloud droplets and ice particles, the chamber walls are up to 10 °C warmer than the enclosed air. This results in  temperature differences of ±1.5 °C in the vertical direction and ±1 °C in the horizontal direction while the air

20 returns to its equilibrium temperature with time constant of about 200 s.

**1 Introduction**

The Intergovernmental Panel on Climate Change (IPCC) considers that the largest source of uncertainty in anthropogenic radiative forcing of the climate is due to increased aerosol since pre-industrial times, and its effect on clouds (Myhre et al., 2013). Most of the increased aerosol has resulted from anthropogenic precursor vapours that, after oxidation in the atmosphere, can form particles which may then grow to become new cloud condensation nuclei (CCN). By current estimates,  more than half of all CCN originate from nucleation rather than being emitted directly into the atmosphere  (Gordon et al., 2017), but the vapours and mechanisms responsible remain relatively poorly known.

~~A solution for studying the previously refered processes is the use of experimental chambers inside which a thorough control of all important parameters is ensured (Paulsen et al., 2005; Zink, 2002). Temperature is an important parameter in atmospheric process estimation (Gordon et al., 2016). Maintaining temperature stability and uniformity in these chamber measurements ensures that the chemical reaction rates in the chamber do no fluctuate either in time or in space (Maahs, 1983). Accurate measurement of temperature is also necessary to measure the onset of ice formation in chamber experiments (Riechers et al., 2013; Connolly et al., 2012)~~
[revised manuscript text omitted]

Opsens Inc.: Quebec, Canada, https://opsens.com.

Paulsen, D., Dommen, J., Kalberer, M., Prévôt, A. S. H., Richter, R., Sax, M., Steinbacher, M., Weingartner, E., and Baltensperger, U.: Secondary Organic Aerosol Formation by Irradiation of 1, 3, 5-Trimethylbenzene-NOx-H2O in a New Reaction Chamber for Atmospheric Chemistry and Physics, Environmental Science & Technology, 39, 2668–2678, doi:10.1021/es0489137, https://doi.org/10.1021/es0489137, 2005.

Raes, F. and Janssens, A.: Ion-induced aerosol formation in a H2O-H2SO4 system—II. Numerical calculations and conclusions, Journal of Aerosol Science, 17, 715–722, doi:10.1016/0021-8502(86)90051-0, https://doi.org/10.1016/0021-8502(86)90051-0, 1986.

Riechers, B., Wittbracht, F., Hütten, A., and Koop, T.: The homogeneous ice nucleation rate of water droplets produced in a microfluidic device and the role of temperature uncertainty, Physical Chemistry Chemical Physics, 15, 5873, doi:10.1039/c3cp42437e, https://doi.org/10.1039/c3cp42437e, 2013.

Schnitzhofer, R., Metzger, A., Breitenlechner, M., Jud, W., Heinritzi, M., De Menezes, L.-P., Duplissy, J., Guida, R., Haider, S., Kirkby, J., Mathot, S., Minginette, P., Onnela, A., Walther, H., Wasem, A., and Hansel, A.: Characterisation of organic contaminants in the CLOUD chamber at CERN, Atmospheric Measurement Techniques, 7, 2159–2168, doi:10.5194/amt-7-2159-2014, https://www.atmos-meas-tech.net/7/2159/2014/, 2014.

Suller, V. P. and Petit-Jean-Genaz, C., eds.: EPAC 94. Proceedings, 4th European Particle Accelerator Conference, London, UK, June 27 - July 1, 1994. Vol. 1-3, 1995.

Voigtlander, J., Duplissy, J., Rondo, L., Kürten, A., and Stratmann, F.: Numerical simulations of mixing conditions and aerosol dynamics in the CERN CLOUD chamber, Atmospheric Chemistry and Physics, 12, 2205–2214, doi:10.5194/acp-12-2205-2012, http://www.atmos-chem-phys.net/12/2205/2012/, 2012.

WIKA Alexander Wiegand SE & Co. KG: 63911 Klingenberg, Germany, http://en-co.wika.de/home_en_co. WIKA.

Wilson, C. T. R. and Wilson, J. G.: On the Falling Cloud-Chamber and on a Radial-Expansion Chamber, Proceedings of the Royal Society A: Mathematical, Physical and Engineering Sciences, 148, 523–533, doi:10.1098/rspa.1935.0032, https://doi.org/10.1098/rspa.1935.0032, 1935.

Zink, P.: Cryo-chamber simulation of stratospheric H2SO4/H2O particles: Composition analysis and model comparison, Geophysical Research Letters, 29, doi:10.1029/2001gl013296, https://doi.org/10.1029/2001gl013296, 2002.

[revised manuscript text omitted]